# A Provably Convergent and Practical Algorithm for Min-Max Optimization with Applications to GANs

## Abstract

We present a first-order algorithm for nonconvex-nonconcave min-max optimization problems such as those that arise in training GANs. Our algorithm provably converges in $\mathrm{poly}(d, L, b)$ steps for any loss function $f : \mathbb{R}^d \times \mathbb{R}^d \to \mathbb{R}$ which is $b$-bounded with $L$-Lipschitz gradient. To achieve convergence, we 1) give a novel approximation to the global strategy of the max-player based on first-order algorithms such as gradient ascent, and 2) empower the min-player to look ahead and simulate the max-player's response for arbitrarily many steps, but restrict the min-player to move according to updates sampled from a stochastic gradient oracle. Our algorithm, when used to train GANs on synthetic and real-world datasets, does not cycle, results in GANs that seem to avoid mode collapse, and achieves a training time per iteration and memory requirement similar to gradient descent-ascent.

## 1 Introduction

We consider the problem of min-max optimization $\min_{x \in \mathbb{R}^d} \max_{y \in \mathbb{R}^d} f(x, y)$, where the loss function $f$ may be nonconvex in $x$ and nonconcave in $y$. Min-max optimization of such loss functions has many applications to machine learning, including to GANs (Goodfellow et al., 2014) and adversarial training (Madry et al., 2018). In particular, following Goodfellow et al. (2014), GAN training can be formulated as a min-max optimization problem where $x$ encodes the parameters of a "generator" network, and $y$ encodes the parameters of a "discriminator" network. Unlike standard minimization problems, the min-max nature of GANs makes them particularly difficult to train (Goodfellow, 2017), and has received wide attention. A common algorithm to solve these min-max optimization problems, gradient descent ascent (GDA), alternates between stochastic gradient descent steps for $x$ and ascent steps for $y$.[1] The advantage of GDA is that it just requires first-order access to $f$ and each iteration is efficient in terms of memory and time, making it quite practical. However, as many works have observed, GDA can suffer from issues such as cycling (Arjovsky & Bottou, 2017) and "mode collapse" (Dumoulin et al., 2017; Che et al., 2017; Santurkar et al., 2018).

Several recent works have focused on finding convergent first-order algorithms for min-max optimization (Rafique et al., 2018; Daskalakis et al., 2018; Liang & Stokes, 2019; Gidel et al., 2019b; Mertikopoulos et al., 2019; Nouiehed et al., 2019; Lu et al., 2020; Lin et al., 2020; Mokhtari et al., 2019; Thekumparampil et al., 2019; Mokhtari et al., 2020). However, these algorithms are also not guaranteed to converge for general nonconvex-nonconcave min-max problems. The challenge is that min-max optimization generalizes nonconvex minimization, which, in general, is intractable. Algorithms for nonconvex minimization resort to finding "local" optima or assume a starting point "close" to a global optimum. However, unlike minimization problems where local notions of optima exist (Nesterov & Polyak, 2006), it has been challenging to define a notion of convergent points for min-max optimization, and most notions of local optima considered in previous works (Daskalakis & Panageas, 2018; Jin et al., 2020; Fiez et al., 2019) require significant restrictions for existence.

**Our contributions.** Our main result is a new first-order algorithm for min-max optimization (Algorithm 1) that for any $\varepsilon > 0$, any nonconvex-nonconcave loss function, and *any* starting point, converges in $\mathrm{poly}(d, L, b, 1/\varepsilon)$ steps, if $f$ is $b$-bounded with $L$-Lipschitz gradient (Theorem 2.3).

---

[1] In practice, gradients steps are often replaced by ADAM steps; we ignore this distinction for this discussion.

A key ingredient in our result is an approximation to the global max function $\max_{z \in \mathbb{R}^d} f(x, z)$. Unlike GDA and related algorithms that alternate between updating the discriminator and generator in an incremental fashion, our algorithm lets the discriminator run a convergent algorithm (such as gradient ascent) *until it reaches a first-order stationary point.* We then empower the generator to simulate the discriminator's response for arbitrarily many gradient ascent updates. Roughly, at each iteration of our algorithm, the min-player proposes a stochastic (batch) gradient update for $x$ and simulates the response of the max-player with gradient ascent steps for $y$ until it reaches a first-order stationary point. If the resulting loss has decreased, the updates for $x$ and $y$ are accepted; otherwise they are only accepted with a small probability (*a la* simulated annealing).

The point $(x^\star, y^\star)$ returned by our algorithm satisfies the following guarantee: if the min-player proposes a stochastic gradient descent update to $x^\star$, and the max-player is allowed to respond by updating $y^\star$ using *any* "path" that increases the loss at a rate of at least $\varepsilon$ — with high probability, the final loss cannot decrease by more than $\varepsilon$. See Section 2 for our convergence guarantees, Section 4 for the key ideas in our proof, and Appendix C for a comparison to previous notions of convergence.

Empirically, we apply our algorithm for training GANs (with the cross-entropy loss) on both synthetic (mixture of Gaussians) and real-world (MNIST and CIFAR-10) datasets (Section 3). We compare our algorithm's performance against two related algorithms: gradient/ADAM descent ascent (with one or multiple discriminator steps), and Unrolled GANs (Metz et al., 2017). Our simulations with MNIST (Figure 1) and mixture of Gaussians (Figure 2) indicate that training GANs using our algorithm can avoid mode collapse and cycling. For instance, on the Gaussian mixture dataset, we found that by around the 1500'th iteration GDA learned only one mode in $100\%$ of the runs, and cycled between multiple modes. In contrast, our algorithm learned all four modes in $68\%$ of the runs, and three modes in $26\%$ of the runs. On 0-1 MNIST, we found that GDA tends to briefly generate shapes that look like a combination of 0's and 1's, then switches between generating only 1's and only 0's. In contrast, our algorithm seems to learn to generate both 0's and 1's early on and does not stop generating either digit. GANs trained using our algorithm generated both digits by the 1000'th iteration in $86\%$ of the runs, while those trained using GDA only did so in $23\%$ of the runs. Our CIFAR-10 simulations (Figure 3) indicate that our algorithm trains more stably, resulting in a lower mean and standard deviation for FID scores compared to GDA. Furthermore, the per-step computational and memory cost of our algorithm is similar to GDA indicating that our algorithm can scale to larger datasets.

**Related work**

*Guaranteed convergence for min-max optimization.* Several works have studied GDA dynamics in GANs (Nagarajan & Kolter, 2017; Mescheder et al., 2017; Li et al., 2018; Balduzzi et al., 2018; Daskalakis & Panageas, 2018; Jin et al., 2020) and established that GDA suffers from severe limitations: GDA can exhibit rotation around some points, or otherwise fail to converge. Thus, we cannot expect global convergence guarantees for GDA. To address these convergence issues for GDA, multiple works have proposed algorithms based on Optimistic Mirror Descent (OMD), Extra-gradient method, or similar approaches (Gidel et al., 2019b; Daskalakis et al., 2018; Liang & Stokes, 2019; Daskalakis & Panageas, 2019; Mokhtari et al., 2019; 2020). These algorithms avoid some of the pathological behaviors of GDA and achieve guaranteed convergence in $\text{poly}(\kappa, \log(1/\varepsilon))$ iterations where $\kappa$ is the condition number of $f$. However, all these results either require convexity/concavity assumptions on $f$, which usually do not hold for GANs, or require that the starting point lies in a small region around an equilibrium point, and hence provide no guarantees for an arbitrary initialization. Some works also provide convergence guarantees for min-max optimization (Nemirovski & Yudin, 1978; Kinderlehrer & Stampacchia, 1980; Nemirovski, 2004; Rafique et al., 2018; Lu et al., 2020; Lin et al., 2020; Nouiehed et al., 2019; Thekumparampil et al., 2019). However, they require $f$ to be concave in $y$, again limiting their applicability.

As for nonconvex-nonconcave min-max optimization, Heusel et al. (2017) prove convergence of finite-step GDA, under the assumption that the underlying continuous dynamics converge to a local min-max optimum (this assumption may not even hold for $f$ that is bi-linear). Jin et al. (2020) present a version of GDA for min-max optimization (generalized by Fiez et al. (2019)) such that if the algorithm converges, the convergence point is a local min-max optimum. Both these results require that the min-player use a vanishingly small step size relative to the max-player, resulting in slow convergence. Wang et al. (2020) present an algorithm that can converge for nonconvex-nonconcave functions, but requires the initial point to lie in a region close a local min-max optimum (such optima are not guaranteed to exist). In contrast to the above works, our algorithm is guaranteed to

converge for any nonconvex-nonconcave loss, from any starting point, in $\mathrm{poly}(d, L, b, 1/\varepsilon)$ steps, if $f$ is $b$-bounded with $L$-Lipschitz gradient.

*Greedy paths.* The paths along which the max-player is allowed to make updates in our equilibrium definition are inspired from the work of Mangoubi & Vishnoi (2020), which gives a second-order algorithm for min-max optimization. The "greedy paths" considered in their work are defined such that at every point along these paths, $f$ is non-decreasing, and the first derivative of $f$ is at least $\varepsilon$ or the 2nd derivative is at least $\sqrt{\varepsilon}$. In contrast, we just require a condition on the first derivative of $f$ along the path. This distinction gives rise to a different notion of equilibrium than the one presented in their work. The first-order condition on the paths crucially also results in our algorithm being applicable to machine learning settings where only first-order oracles are available, because unlike Mangoubi & Vishnoi (2020), traversing such a path only requires first-order access to $f$.

*Training GANs.* Starting with Goodfellow et al. (2014), there has been considerable work to develop algorithms to train GANs. One line of work focuses on modifying the loss to improve convergence (Arjovsky et al., 2017; Bellemare et al., 2017; Lim & Ye, 2017; Mao et al., 2017; Salimans et al., 2018; Metz et al., 2017). Another line of work regularizes the discriminator using gradient penalties or spectral normalization (Gulrajani et al., 2017; Kodali et al., 2017; Miyato et al., 2018).

Metz et al. (2017) introduced Unrolled GANs, where the generator optimizes an "unrolled" loss function that allows the generator to simulate a fixed number of discriminator updates. While this has some similarity to our algorithm there are two important distinctions: 1) the discriminator in Unrolled GANs may not reach a first-order stationary point, and hence their algorithm does not come with any convergence guarantees, and 2) unlike our algorithm, the implementation of the generator in Unrolled GANs requires memory that grows with the number of discriminator steps, limiting its scalability. We observe that our algorithm, applied to training GANs, trains stably and avoids mode collapse, while achieving a training time per iteration and memory requirements that are similar to GDA, and much lower than Unrolled GANs (Metz et al., 2017) (see also the discussion in Section 5).

**Remark 1.1** (Loss functions in GANs). *Loss functions $f : \mathbb{R}^d \times \mathbb{R}^d \to \mathbb{R}$ which take bounded values on $\mathbb{R}^d \times \mathbb{R}^d$ arise in many GAN applications. For instance, GANs with mean-squared error loss Mao et al. (2017) have uniformly bounded $f$. GANs with cross entropy loss Goodfellow et al. (2014) have $f$ uniformly bounded above, and Wasserstein GANs Arjovsky et al. (2017) have a loss function $f(x, y)$ which is bounded above as a function of $y$ and is uniformly bounded below.*

## 2 THEORETICAL RESULTS

We consider the problem $\min_x \max_y f(x, y)$, where $x, y \in \mathbb{R}^d$, and $f$ is a function $\mathbb{R}^d \times \mathbb{R}^d \to \mathbb{R}$. We consider $f$ that is an empirical risk loss over $m$ training examples. Thus, we have $f = \frac{1}{m} \sum_{i \in [m]} f_i$, and are given access to $f$ via a randomized oracle $F$ such that $\mathbb{E}[F] = f$. We call such an oracle a stochastic zeroth-order oracle for $f$. We are also given randomized oracles $G_x, G_y$ for $\nabla_x f, \nabla_y f$, such that $\mathbb{E}[G_x] = \nabla_x f$, and $\mathbb{E}[G_y] = \nabla_y f$. We call such oracles stochastic gradient oracles for $f$. In practice, these oracles are computed by randomly sampling a "batch" $B \subseteq [m]$ and returning $F = 1/|B| \sum_{i \in B} f_i$, $G_x = 1/|B| \sum_{i \in B} \nabla_x f_i$, and $G_y = 1/|B| \sum_{i \in B} \nabla_y f_i$.

For our convergence guarantees, we require bounds on standard smoothness parameters for functions $f_i$ : $b$ such that for all $i$ and all $x, y$, we have $|f_i(x, y)| \leq b$, and $L$ such that $\|\nabla f_i(x, y) - \nabla f_i(x', y')\|_2 \leq L\|x - x'\|_2 + L\|y - y'\|_2$. Such smoothness/Lipschitz bounds are standard in convergence guarantees for optimization algorithms (Bubeck, 2017; Nesterov & Polyak, 2006; Ge et al., 2015), and imply that $f$ is also continuous, $b$-bounded, and $L$-gradient-Lipschitz.

Our algorithm is described informally in Algorithm 1 and formally as Algorithm 2 in the Appendix.

**Intuition for the algorithm.** To solve the min-max problem, the max-player would ideally find the global maximum $\max_z f(x, z)$. However, since $f$ may be nonconcave in $y$, finding the global maximum may be computationally intractable. To get around this problem, roughly speaking, in our algorithm the max-player computes its update by running gradient ascent until it reaches a first-order $\varepsilon$-stationary point $y'$, that is, a point where $\|\nabla_y f(x, y')\| \leq \varepsilon$. This allows our algorithm to compute an approximation $\mathcal{L}_\varepsilon(x, y) = f(x, y')$ for the global maximum. (Note that even though $\max_z f(x, z)$ is only a function of $x$, $\mathcal{L}_\varepsilon$ may depend on both $x$ and the initial point $y$.)

---

**Algorithm 1** Algorithm for min-max optimization

---

**input:** A stochastic zeroth-order oracle $F$ for loss function $f : \mathbb{R}^d \times \mathbb{R}^d \to \mathbb{R}$, and stochastic gradient oracles $G_x$ for $\nabla_x f$, and $G_y$ for $\nabla_y f$. An initial point $(x, y)$, and an error parameter $\varepsilon$.
**output:** A point $(x^\star, y^\star)$
**hyperparameters:** $r_{\max}$ (maximum number of rejections); $\tau_1$ (hyperparameters for annealing); ADAM hyperparameters

> Set $r \leftarrow 0, i \leftarrow 0$
> **while** $r \leq r_{\max}$ **do**
> > $f_{\text{old}} \leftarrow F(x, y), \quad i \leftarrow i + 1$
> > Set $G_x \leftarrow G_x(x, y)$ {*Compute a stochastic gradient*}
> > Use stochastic gradient $G_x$ to compute a one-step ADAM update $\Delta$ for $x$
> > Set $x' \leftarrow x + \Delta$ {*Compute the proposed update for the min-player*}
> > Starting at point $y$, use stochastic gradients $G_y(x', \cdot)$ to run multiple ADAM steps in the $y$-variable, until a point $y'$ is reached such that $\|G_y(x', y')\|_1 \leq \varepsilon$ {*Simulate max-player's update*}
> > Set $f_{\text{new}} \leftarrow F(x', y')$ {*Compute the new loss value*}
> > Set Accept $\leftarrow$ True.
> > **if** $f_{\text{new}} > f_{\text{old}} - \varepsilon/2$, set Accept $\leftarrow$ False with probability $\max(0, 1 - e^{-i/\tau_1})$ {*accept or reject*}
> > **if** Accept = True **then** Set $x \leftarrow x', y \leftarrow y', r \leftarrow 0$ {*Accept the updates*}
> > **else** Set $r \leftarrow r + 1$ {*Reject the updates, and track how many successive steps were rejected.*}
> **return** $(x, y)$

---

We would like the min-player to minimize $\mathcal{L}_\varepsilon(x, y)$. Ideally, the min-player would make updates in the direction $-\nabla_x \mathcal{L}_\varepsilon$. However, $\mathcal{L}_\varepsilon(x, y)$ may not be differentiable and may even be discontinuous in $x$ (see Section 2.2 for an example), making it challenging to optimize. Moreover, even at points where $\mathcal{L}_\varepsilon$ is differentiable, computing $\nabla_x \mathcal{L}_\varepsilon$ may require memory proportional to the number of max-player steps used to compute $\mathcal{L}_\varepsilon$ (for instance, this is the case for Unrolled GANs (Metz et al., 2017)). For this reason, we only provide our min-player with access to the value of $\mathcal{L}_\varepsilon$.

One approach to minimize $\mathcal{L}_\varepsilon$ would be to use a zeroth-order optimization procedure where the min-player proposes a random update to $x$, and then only accept this update if it results in a decrease in $\mathcal{L}_\varepsilon$. At each iteration of our algorithm, the min-player proposes an update roughly in the direction $-\nabla_x f(x, y)$. To motivate this choice, note that once the min-player proposes an update $\Delta$ to $x$, the max-player's updates will only increase $f$, *i.e.*, $\mathcal{L}_\varepsilon(x + \Delta, y) \geq f(x + \Delta, y)$. Moreover, since $y$ is a first-order stationary point of $f(x, \cdot)$ (because $y$ was computed using gradient ascent in the *previous* iteration), we also have $\mathcal{L}_\varepsilon(x, y) = f(x, y)$. Therefore, we want an update $\Delta$ such that

$$f(x + \Delta, y) \leq \mathcal{L}_\varepsilon(x + \Delta, y) \leq \mathcal{L}_\varepsilon(x, y) = f(x, y), \tag{1}$$

which implies that any proposed step which decreases $\mathcal{L}_\varepsilon$ must also decrease $f$ (although the converse is not true). This motivates proposing steps in the direction of the gradient $-\nabla_x f(x, y)$.

Unfortunately, updates in the direction $-\nabla_x f(x, y)$ do not necessarily decrease $\mathcal{L}_\varepsilon$. Our algorithm instead has the min-player perform a random search by proposing a stochastic update in the direction of a batch gradient with mean $-\nabla_x f(x, y)$ (or, more precisely, the ADAM batch gradients), and accepts this update only if $\mathcal{L}_\varepsilon$ decreases by some fixed amount. We show empirically that these directions allow the algorithm to rapidly decrease the simulated loss. The fact that $\mathcal{L}_\varepsilon$ decreases whenever the min-player takes a step allows us to guarantee that our algorithm eventually converges.

A final issue is that converging to a *local* minimum point does not guarantee that the point is desirable from an applications standpoint. To allow our algorithm to escape undesirable local minima of $\mathcal{L}_\varepsilon(\cdot, y)$, we use a randomized accept-reject rule inspired by simulated annealing – if the resulting loss has decreased the updates for $x$ and $y$ are accepted; otherwise they are only accepted with a small probability $e^{-i/\tau_1}$, where $\tau_1$ is a "temperature" parameter.

## 2.1 CONVERGENCE GUARANTEES

We first formally define "simulated loss" and what it means for $f$ to increase rapidly.

**Definition 2.1.** *For any $x, y$, and $\varepsilon > 0$, define $\mathcal{E}(\varepsilon, x, y) \subseteq \mathbb{R}^d$ to be points $w$ s.t. there is a continuous and (except at finitely many points) differentiable path $\gamma(t)$ starting at $y$, ending at $w$,*

*and moving with "speed" at most 1 in the $\ell_\infty$-norm [2] $\left\| \frac{\mathrm{d}}{\mathrm{d}t}\gamma(t)\right\|_\infty \leq 1$ such that at any point on $\gamma$, [3]*

$$\frac{\mathrm{d}}{\mathrm{d}t}f(x,\gamma(t)) > \varepsilon. \tag{2}$$

*We define $\mathcal{L}_\varepsilon(x,y) := \sup_{w\in\mathcal{E}(\varepsilon,x,y)} f(x,w)$, and refer to it as the simulated loss.*

A few remarks are in order. Observe that $\mathcal{L}_{-\infty}(x,y) = \max_y f(x,y)$. Further, if $\|\nabla_y f(x,y)\|_1 \leq \varepsilon$, then $\mathcal{E}(\varepsilon,x,y) = \{y\}$ and $\mathcal{L}_\varepsilon(x,y) = f(x,y)$ (this follows from Hölder's inequality, since $\gamma(t)$ has speed at most 1 in the $\ell_\infty$-norm, the dual of the $\ell_1$-norm). Note that the path $\gamma$ need not be in the direction of the gradient, and there can potentially be infinitely many such paths starting at $y$.

Unfortunately, the simulated loss may not even be continuous[4] in $x$, and thus, gradient-based notions of approximate local minima do not apply. To bypass this discontinuity (and hence non-differentiability), we use the idea to sample updates to $x$, and test whether $\mathcal{L}_\varepsilon$ has decreased (equation 35). This leads to the following definition of local min-max equilibrium (see also Section 2.3):

**Definition 2.2.** *Given a distribution $\mathcal{D}_{x,y}$ for each $x, y \in \mathbb{R}^d$ and $\varepsilon^\star > 0$, we say that $(x^\star, y^\star)$ is an $\varepsilon^\star$-local min-max equilibrium with respect to the distribution $\mathcal{D}$ if*

$$\|\nabla_y f(x^\star, y^\star)\|_1 \leq \varepsilon^\star, \quad and \quad , \tag{3}$$
$$\Pr_{\Delta\sim\mathcal{D}_{x^\star,y^\star}}\left[\mathcal{L}_{\varepsilon^\star}(x^\star + \Delta, y^\star) < \mathcal{L}_{\varepsilon^\star}(x^\star, y^\star) - \varepsilon^\star\right] < \varepsilon^\star, \tag{4}$$

In our main result, we set $\mathcal{D}$ to roughly be the distribution of ADAM stochastic gradients for $-\nabla_x f$. Also note that from the above discussion, equation 34 implies that $\mathcal{L}_{\varepsilon^\star}(x^\star, y^\star) = f(x^\star, y^\star)$. To allow convergence of ADAM (and avoid non-convergence issues such as those encountered in Reddi et al. (2019)), in our main result we constrain the values of the ADAM hyperparameters [5]. Now, we can state the formal guarantees of our algorithm.

**Theorem 2.3.** *Algorithm 2, with appropriate hyperparameters for ADAM and some constant $\tau_1 > 0$, given access to stochastic zeroth-order and gradient oracles for a function $f = \sum_{i\in[m]} f_i$ where each $f_i$ is $b$-bounded with $L$-Lipschitz gradient for some $b, L > 0$, and $\varepsilon > 0$, with probability at least $9/10$ returns a point $(x^\star, y^\star) \in \mathbb{R}^d \times \mathbb{R}^d$ such that, for some $\varepsilon^\star \in [\frac{1}{2}\varepsilon, \varepsilon]$, the point $(x^\star, y^\star)$ is an $\varepsilon^\star$-local min-max equilibrium point with respect to the distribution $\mathcal{D}_{x,y}$, where $\mathcal{D}_{x,y}$ is the distribution of $G_x(x,y)/\sqrt{G_x^2(x,y)}$, where the division "/" is applied element-wise. The number of stochastic gradient and function evaluations required by the algorithm is $\mathrm{poly}(1/\varepsilon, d, b, L)$.*

We present key ideas in the proof in Section 4, and a proof overview in Section B. The full proof appears in Appendix D. Note that $\mathcal{D}_{x,y}$ is the distribution of the stochastic gradient updates with element-wise normalizations. Roughly, this corresponds to the distribution of ADAM steps taken by the algorithm for updating $x$ if one uses a small step-size parameter for ADAM.

We conclude this section with a few technical remarks about the theorem. Our algorithm could provide guarantees with respect to other distributions $\mathcal{D}_{x,y}$ in equation 35 by sampling update steps for $x$ from $\mathcal{D}_{x,y}$ instead of ADAM. The norm in the guarantees of the stationary point $y^\star$ for our algorithm is $\ell_1$ since we use ADAM for updating $y$. A simpler version of this algorithm using SGD would result in an $\ell_2$-norm guarantee.

**Comparison to notions of local optimality.** Definition 2.2 provides a new notion of equilibrium for min-max optimization. Consider the problem $\min_x \max_y f(x,y)$, but with constraints on the player's updates to $x, y$ — the max player is restricted to updating $y$ via a path which increases $f(x,y)$ at a rate of at least $\varepsilon^\star$ at every point on the path, and the min player proposes an update for $x$, $\Delta$ sampled from $\mathcal{D}$. Then $(x^\star, y^\star)$ satisfies Definition 2.2 if and only if (i) there is no update to $y^\star$ that the max player can make which will increase the loss at a rate of at least $\varepsilon^\star$ (equation 34), and, (ii) with probability at least $1 - \varepsilon^\star$ for a random step $\Delta \sim \mathcal{D}$ proposed by the min player, the above-constrained max player can update $y^\star$ s.t. the overall decrease in the loss is at most $\varepsilon^\star$ from its original value $f(x^\star, y^\star)$.

---

[2]We use the $\ell_\infty$-norm in place of the Euclidean $\ell_2$-norm, as it is a more natural norm for ADAM gradients.
[3]In this equation the derivative $\frac{\mathrm{d}}{\mathrm{d}t}$ is taken from the right.
[4]Consider the example $f(x,y) = \min(x^2 y^2, 1)$. The simulated loss function for $\varepsilon > 0$ is $\mathcal{L}_\varepsilon(x,y) = f(x,y)$ if $2x^2 y < \varepsilon$, and 1 otherwise. Thus $\mathcal{L}_{1/2}$ is discontinuous at $(1/2, 1)$.
[5]In particular, we set the ADAM hyperparameters $\beta_1, \beta_2$ to be $\beta_1 = \beta_2 = 0$.

As one comparison to a previous notion of local optimality, any point which is a local optimum under the definition used previously e.g. in Daskalakis & Panageas (2018), also satisfies our Definition 2.2 for small enough $\varepsilon$ and distribution $\mathcal{D}$ corresponding to small-enough step size. On the other hand, previous notions of local optima including the one in Daskalakis & Panageas (2018) are not guaranteed to exist in a general setting, unlike our definition. (See Appendix C for a detailed comparison of how Definition 2.2 relates to previously proposed notions of local optimality).

## 3 EMPIRICAL RESULTS

We seek to apply our min-max optimization algorithm for training GANs on both real-world and synthetic datasets. Following Goodfellow et al. (2014), we formulate GAN training as a min-max optimization problem using the cross entropy loss, $f(x, y) = \log(\mathcal{D}_y(\zeta)) + \log(1 - \mathcal{D}_y(\mathcal{G}_x(\xi)))$, where $x, y$ are the weights of the generator and discriminator networks $\mathcal{G}$ and $\mathcal{D}$ respectively, $\zeta$ is sampled from the data, and $\xi \sim N(0, I_d)$. For this loss, the smoothness parameters $b, L$ may not be finite. To adapt Alg. 1 to training GANs, we make the following simplifications in our simulations:

**(1)** Temperature schedule: We use a fixed temperature $\tau$, constant with iteration $i$, making it simpler to choose a good temperature value rather than a temperature schedule. **(2)** Accept/reject rule: We replace the randomized acceptance rule with a deterministic rule: If $f_{\text{new}} \leq f_{\text{old}}$ we accept the proposed step, and if $f_{\text{new}} > f_{\text{old}}$ we only accept if $i$ is a multiple of $e^{1/\tau}$, corresponding to an average acceptance rate of $e^{-1/\tau}$. **(3)** Discriminator steps: We take a fixed number of discriminator steps at each iteration, instead of taking as many steps needed to achieve a small gradient.

These simplifications do not seem to significantly affect our algorithm's performance (see Appendix F.5 for simulations showing it effectively trains GANs without most of these simplifications). Moreover, our simulations show a smaller number of discriminator steps $k$ is usually sufficient in practice.

**Datasets and Metrics.** We perform simulations on MNIST (LeCun et al., 2010) and CIFAR-10 (Krizhevsky et al.) datasets to evaluate whether GANs trained using our algorithm converge, and whether they are able to learn the target distribution. Convergence is evaluated by visual inspection (for MNIST and CIFAR), and by tracking the loss (for MNIST) and FID scores (for CIFAR).

As noted by previous works (Borji, 2019; Metz et al., 2017; Srivastava et al., 2017), it is challenging to detect mode collapse on CIFAR and MNIST, visually or using standard quantitative metrics such as FID scores, because CIFAR (and to some extent MNIST) do not have well-separated modes. Thus, we consider two datasets, one real and one synthetic, with well-separated modes, whence mode collapse can be clearly detected by visual inspection.

For the real dataset we consider the **0-1 MNIST** dataset (MNIST restricted to digits labeled 0 or 1). The synthetic dataset consists of 512 points sampled from a mixture of **four Gaussians** in two dimensions with standard deviation 0.01 and means at $(0, 1)$, $(1, 0)$, $(-1, 0)$ and $(0, -1)$.

**Hyperparameters and hardware.** The details of the networks and hyperparameter choices are given in Appendix E. Simulations on MNIST and Gaussian datasets used four 3.0 GHz Intel Scalable CPUs, provided by AWS. On CIFAR-10, we used one High freq. Intel Xeon E5-2686 v4 GPU.

### 3.1 EVALUATING THE PERFORMANCE OF OUR ALGORITHM

We compare our algorithm's performance to both GDA and unrolled GANs. All algorithms are implemented using ADAM (Kingma & Ba, 2015).

**MNIST.** We trained a GAN on the full MNIST dataset using our algorithm for 39,000 iterations (with $k = 1$ discriminator steps and acceptance rate $e^{-1/\tau} = 1/5$). We ran this simulation five times; each time the GAN learned to generate all ten digits (see Appendix F.1 for generated images).

**0-1 MNIST.** We trained a GAN using our algorithm on the 0-1 MNIST dataset for 30,000 iterations and ploted a moving average of the loss values. We repeated this simulation five times; in each of the five runs our algorithm learned to generate digits which look like both the "0" and "1" digits, and the loss seems to decrease and stabilize once our algorithm learns how to generate the two digits. (See Appendix F.3 for the generated images and loss value plots.)

GDA                                    Our algorithm

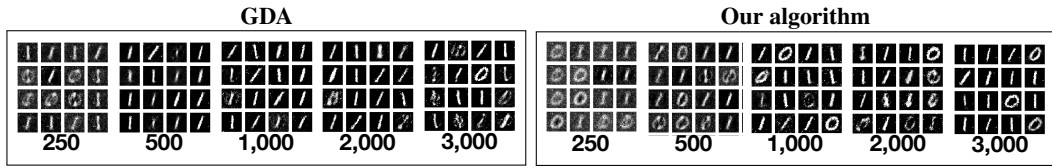

Figure 1: We trained a GAN using our algorithm on 0-1 MNIST for 30,000 iterations (with $k = 1$ discriminator steps and acceptance rate $e^{-1/\tau} = 1/5$). We repeated this experiment 22 times for our algorithm and 13 times for GDA. Shown here are the images generated from one of these runs at various iterations for our algorithm (right) and GDA (left) (see also Appendix F.3 for images from other runs).

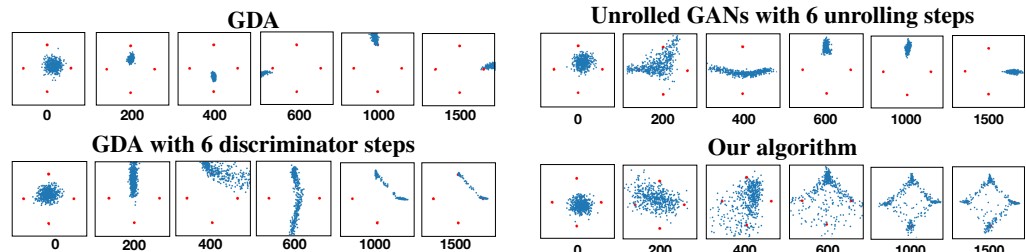

Figure 2: Our algorithm (bottom right), unrolled GANs with $k = 6$ unrolling steps (top right), and GDA with $k = 1$ (top left) and $k = 6$ discriminator steps (bottom left). Each algorithm was trained on a 4-Gaussian mixture for 1500 iterations. Our algorithm used $k = 6$ discriminator steps and acceptance rate $e^{-1/\tau} = 1/4$. Plots show the points generated by each of these algorithms after the specified number of iterations.

**CIFAR-10.** We ran our algorithm (with $k = 1$ discriminator steps and acceptance rate $e^{-1/\tau} = 1/2$) on CIFAR-10 for 50,000 iterations. We compare with GDA with $k = 1$ discriminator steps. We plotted the FID scores for both algorithms. We found that both algorithms have similar FID scores which decrease over time, and produce images of similar quality after 50,000 iterations (Figure 3).

**Clock time per iteration.** When training on the 0-1 MNIST dataset (with $k = 1$ discriminator steps each iteration), our algorithm took 1.4 seconds per iteration on the AWS CPU server. On the same machine, GDA took 0.85 seconds per iteration. When training on CIFAR-10, our algorithm and GDA both took the same amount of time per iteration, 0.08 seconds, on the AWS GPU server.

**Mitigating mode-collapse on 0-1 MNIST.** We trained GANs using both GDA and our algorithm on the 0-1 MNIST dataset, and ran each algorithm for 3000 iterations (Figure 1). GDA seems to briefly generate shapes that look like a combination of 0's and 1's, then switches to generating only 1's, and then re-learns how to generate 0's. In contrast, our algorithm seems to learn how to generate both 0's and 1's early on and does not mode collapse to either digit.

We repeated this simulation 22 times for our algorithm and 13 times for GDA, and visually inspected the images at iteration 1000. GANs trained using our algorithm generated both digits by the 1000'th iteration in 86% of the runs, while those trained using GDA only did so in 23% of the runs at the 1000'th iteration (see Appendix F.4 for images from all runs).

**Mitigating mode-collapse on synthetic data.** We trained GANs on the 4-Gaussian mixture dataset for 1500 iterations (Figure 2) using our algorithm, unrolled GANs with $k = 6$ unrolling steps, and GDA with $k = 1$ and $k = 6$ discriminator steps. We repeated each simulation 10-20 times.

By the 1500'th iteration GDA with $k = 1$ discriminator steps seems to have learned only one mode in 100% of the runs. GDA with $k = 6$ discriminator steps learned two modes 65% of the runs, one mode 20% of runs, and four modes 15% of runs. Unrolled GANs learned one mode 75% of the runs, two modes 15% of the runs, and three modes 10% of the runs. In contrast, our algorithm learned all four modes 68% of the runs, three modes 26% of the runs, and two modes 5% of the runs.

Figure 3: GAN trained using our algorithm (with $k = 1$ discriminator steps and acceptance rate $e^{-1/\tau} = 1/2$) and GDA on CIFAR-10 for 50,000 iterations. The images generated from the resulting generator for both our algorithm (middle) and GDA (left). Over 9 runs, our algorithm achieves a very similar minimum FID score (33.8) compared to GDA (33.0), and a better average FID score over 9 runs (mean $\mu = 35.6$, std. dev. $\sigma = 1.1$) compared to GDA ($\mu = 53.8$, $\sigma = 53.9$). Images are shown from one run each; see Appendix F.2 for full results.

## 4    KEY IDEAS IN THE PROOF

For simplicity, assume $b = L = \tau_1 = 1$. There are two key pieces to proving Theorem 2.3. The first is to show that our algorithm converges to some point $(x^\star, y^\star)$ in $\mathrm{poly}(d, 1/\varepsilon)$ gradient and function evaluations (Lemma D.7). Secondly, we show that, $y^\star$ is a first-order $\varepsilon$-stationary point for $f(x^\star, \cdot)$ and $x^\star$ is, roughly, a $\varepsilon$-local minimum for the simulated loss function $\mathcal{L}_\varepsilon(\cdot, y^\star)$ (Lemma D.9).

**Step1: Bounding the number of gradient evaluations:** After $\Theta(\log(\frac{1}{\varepsilon}))$ steps, the decaying acceptance rate of the simulated annealing step ensures that our algorithm stops whenever $r_{\max} = O(1/\varepsilon)$ proposed steps are rejected in a row. Thus, for every $O(r_{\max}/\varepsilon^2)$ iterations where the algorithm does not terminate, with probability at least $1 - \varepsilon$ the value of the loss decreases by more than $\varepsilon$. Since $f$ is 1-bounded, this implies our algorithm terminates after roughly $O(r_{\max}/\varepsilon^3)$ iterations of the minimization routine (Proposition D.6).

Next, since $f$ is 1-bounded with 1-Lipschitz gradient, in each iteration, we require at most $\mathrm{poly}(d/\varepsilon)$ gradient ascent steps to reach an $\varepsilon$-stationary point. Since each step of the maximization subroutine requires one gradient evaluation, and each iteration of the minimization routine calls the maximization routine exactly once, the total number of gradient evaluations is $\mathrm{poly}(d, 1/\varepsilon)$.

**Step 2: Show $x^\star$ is an $\varepsilon$-local minimum for $\mathcal{L}_\varepsilon(\cdot, y^\star)$ and $y^\star$ is an $\varepsilon$-stationary point.** First, we note that since our algorithm runs the gradient ascent maximization subroutine until it reaches an $\varepsilon$-stationary point, we have, $\|\nabla_{\mathrm{y}} f(x^\star, y^\star)\|_1 \leq \varepsilon$.

Our stopping condition for the algorithm implies the last $r_{\max}$ updates $\Delta$ proposed by the max-player were all rejected, and hence were sampled from the distribution $\mathcal{D}_{x^\star, y^\star}$ of the ADAM gradient at $(x^\star, y^\star)$. Roughly, this implies

$$\Pr_{\Delta \sim \mathcal{D}_{x^\star, y^\star}} \left[ f(x^\star + \Delta, y') \geq f(x^\star, y^\star) - \varepsilon \right] \geq 1 - \varepsilon, \tag{5}$$

where the maximization subroutine computes $y'$ by gradient ascent on $f(x^\star + \Delta, \cdot)$ initialized at $y^\star$. In other words, equation 5 says that at the point $(x^\star, y^\star)$ where our algorithm stops, if the min-player samples an update $\Delta$ from the distribution $\mathcal{D}_{x^\star, y^\star}$, followed by the max-player updating $y^\star$ using gradient ascent, with high probability the final loss value cannot decrease by more than $\varepsilon$.

To show equation 35 holds, we need to replace $f$ in the above equation with the simulated loss $\mathcal{L}_\varepsilon$. We first show that the gradient ascent steps form an "$\varepsilon$-increasing" path, starting at $y^\star$ with endpoint $y'$, along which $f$ increases at rate at least $\varepsilon$ (Prop. D.8). This crucially exploits that our algorithm restricts the max player to only use such "$\varepsilon$-increasing" paths. Since $\mathcal{L}_\varepsilon$ is the supremum of $f$ at the endpoints of *all* such $\varepsilon$-increasing paths starting at $y^\star$, we get

$$f(x^\star + \Delta, y') \leq \mathcal{L}_\varepsilon(x^\star + \Delta, y^\star). \tag{6}$$

Finally, recall from Section 2 that $\|\nabla_{\mathrm{y}} f(x^\star, y^\star)\|_1 \leq \varepsilon^\star$ implies $\mathcal{L}_\varepsilon(x^\star, y^\star) = f(x^\star, y^\star)$. Combining the above observations implies the $\varepsilon$-local minimum condition equation 35.

Note that we could pick any distribution $\mathcal{D}$ for the updates and the proof still holds – the distribution of ADAM gradients works well in practice. Also, we could replace simulated annealing with a deterministic rule, but such an algorithm often gets stuck at poor local equilibria in GAN training.

## 5 CONCLUSION

In this paper, we develop a convergent first-order algorithm for min-max optimization and show how it can lead to a stable and scalable algorithm for training GANs. We prove that our algorithm converges in time polynomial in the dimension and the smoothness parameters of the loss function. Our simulations show that a version of our algorithm can lead to more stable training of GANs on synthetic and real-world datasets. And yet the amount of memory and time required by each iteration of our algorithm is competitive with GDA. Our algorithm synthesizes a first-order approximation to the global strategy of the max-player, a look ahead strategy based on batch gradients for the min-player, and simulated annealing. We believe that these ideas of imposing computational restrictions on the min- and max-players should be useful in obtaining convergent and practical algorithms for other applications of min-max optimization, such as adversarial learning.

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

# A  THE FULL ALGORITHM

---

**Algorithm 2** Algorithm for min-max optimization (formal version)

---

**input:** Stochastic zeroth-order oracle $F$ for loss function $f : \mathbb{R}^d \times \mathbb{R}^d \to \mathbb{R}$, and stochastic gradient oracle $G_x$ for $\nabla_x f$, and $G_y$ for $\nabla_y f$

**input:** Initial point $(x_0, y_0)$, Error parameter $\varepsilon$

**output:** A point $(x^\star, y^\star)$

**hyperparameters:** $r_{\max}$ (max number of rejections); $\tau_1 > 0$ (hyperparameter for simulated annealing); hyperparameters $\alpha, \eta, \delta > 0$ and $\beta_1, \beta_2 \in [0, 1]$ for ADAM

1: Set $i \leftarrow 0$, $r \leftarrow 0$, $a \leftarrow 0$, $m_0 \leftarrow 0$, $v_0 \leftarrow 0$, $\mathsf{m} \leftarrow 0$, $\mathsf{v} \leftarrow 0$, $\mathsf{s} \leftarrow 0$, $\varepsilon_1 = \frac{\varepsilon}{2}$

2: Set $f_{\text{old}} \leftarrow \infty$ {Set $f_{\text{old}}$ to be $\infty$ (or the largest possible value allowed by the computer) to ensure that the first step is accepted}

3: **while** $r \leq r_{\max}$ **do**

4:     Set $i \leftarrow i + 1$

5:     Set $g_{\text{x},i} \leftarrow G_x(x_i, y_i)$ {*Compute proposed stochastic gradient*}

6:     Set $M_{i+1} \leftarrow \beta_1 m_i + (1 - \beta_1) g_{\text{x},i}$, and $V_{i+1} \leftarrow \beta_2 v_i + (1 - \beta_2) g_{\text{x},i}^2$ {*Compute proposed ADAM update for first- and second- moment estimates*}

7:     Set $X_{i+1} \leftarrow x_i - \alpha \frac{1}{1-\beta_1^{a+1}} M_{i+1} / (\sqrt{\frac{1}{1-\beta_2^{a+1}} V_{i+1}} + \delta)$ {*Compute proposed ADAM update of x-variable*}

8:     Run Algorithm 3 with inputs $\mathsf{x} \leftarrow X_{i+1}$, $\mathsf{y}_0 \leftarrow y_i$, $(\mathsf{m}, \mathsf{v}, \mathsf{s})$, and $\varepsilon' \leftarrow \varepsilon_i(1 + \eta)$.        } *Use ADAM optimizer to*

9:     Set $\mathcal{Y}_{i+1} \leftarrow \mathsf{y}_{\text{stationary}}$ and $(\mathsf{M}, \mathsf{V}, \mathsf{S}) \leftarrow (\mathsf{m}_{\text{out}}, \mathsf{v}_{\text{out}}, \mathsf{s}_{\text{out}})$ to be the outputs of Alg. 3.    } *simulate the*

10:    Set $f_{\text{new}} \leftarrow F(X_{i+1}, \mathcal{Y}_{i+1})$        *max player's response.*

11:    **if** $f_{\text{new}} > f_{\text{old}} - \frac{\varepsilon}{4}$, **then**

12:        Set $\mathsf{Accept}_i \leftarrow \mathsf{False}$ with probability $\max(0, 1 - e^{-\frac{i}{\tau_1}})$ {*Decide to accept or reject*}

13:    **if** $\mathsf{Accept}_i = \mathsf{True}$ **then**

14:        Set $x_{i+1} \leftarrow X_{i+1}$, $y_{i+1} \leftarrow \mathcal{Y}_{i+1}$ {accept the proposed x and y updates}

15:        Set $m_{i+1} \leftarrow M_{i+1}$, $v_{i+1} \leftarrow V_{i+1}$, and $(\mathsf{m}, \mathsf{v}, \mathsf{s}) \leftarrow (\mathsf{M}, \mathsf{V}, \mathsf{S})$ {accept the proposal's ADAM moment estimates}

16:        Set $f_{\text{old}} \leftarrow f_{\text{new}}$

17:        Set $a \leftarrow a + 1$ {keep track of how many steps we accepted}

18:        Set $r \leftarrow 0$

19:        Set $\varepsilon_{i+1} \leftarrow \varepsilon_i(1 + \eta)^2$

20:    **else**

21:        Set $x_{i+1} \leftarrow x_i$, $y_{i+1} \leftarrow y_i$ {go back to the previous x and y values}

22:        Set $m_{i+1} \leftarrow m_i$, $v_{i+1} \leftarrow v_i$ {go back to the previous ADAM moment estimates}

23:        $r \leftarrow r + 1$ {Keep track of how many steps were rejected since the last acceptance. If too many steps were rejected in a row, stop the algorithm and output the current weights.}

24:        Set $\varepsilon_{i+1} \leftarrow \varepsilon_i$

25: **return** $(x^\star, y^\star) \leftarrow (x_i, y_i)$

---

---

**Algorithm 3** ADAM (for the max-player updates)

---

**input:** Stochastic gradient oracle $G_y$ for $\nabla_y f$

**input:** $\mathsf{x}, \mathsf{y}_0, \mathsf{m}, \mathsf{v}, \varepsilon'$, number of steps $\mathsf{s}$ taken at previous iterations

**output:** A point $\mathsf{y}_{\text{stationary}}$ which is a first-order $\varepsilon'$-stationary point for $f(\mathsf{x}, \cdot)$, and $\mathsf{s}_{\text{out}}, \mathsf{m}_{\text{out}}, \mathsf{v}_{\text{out}}$

**hyperparameters:** $\eta > 0$; ADAM hyperparameters $\hat{\alpha}, \delta > 0$ and $\beta_1, \beta_2 \in [0, 1]$

1: Set $j \leftarrow 0$
2: Set Stop $=$ False
3: **while** Stop $=$ False **do**
4:     Set $j \leftarrow j + 1$
5:     Set $g_{\mathsf{y},j} \leftarrow G_y(\mathsf{x}, \mathsf{y}_j)$
6:     **if** $\|g_{\mathsf{y},j}\|_1 > \varepsilon'$ **then**
7:         Set $j \leftarrow j + 1$
8:         Set $\mathsf{m}_j \leftarrow \beta_1 \mathsf{m}_j + (1 - \beta_1)g_{\mathsf{y},j}$, and $\mathsf{v}_j \leftarrow \beta_2 \mathsf{v}_j + (1 - \beta_2)g_{\mathsf{y},j}^2$ {*Compute proposed ADAM update for first- and second- moment estimates*}
9:         Set $\mathsf{y}_{j+1} \leftarrow \mathsf{y}_j + \hat{\alpha}\frac{1}{1-\beta_1^{\mathsf{s}+j+1}}\mathsf{m}_j / (\sqrt{\frac{1}{1-\beta_2^{\mathsf{s}+j+1}}\mathsf{v}_j} + \delta)$ {*Compute proposed ADAM update of* $\mathsf{y}$*-variable*}
10:     **else**
11:         Set Stop $=$ True
12: Set $\mathsf{y}_{\text{stationary}} \leftarrow \mathsf{y}_j$ and $\mathsf{s}_{\text{out}} \leftarrow \mathsf{s} + j$, $\mathsf{m}_{\text{out}} \leftarrow \mathsf{m}_j$, $\mathsf{v}_{\text{out}} \leftarrow \mathsf{v}_j$

---

# B  PROOF OVERVIEW

To prove Theorem 2.3 we would like to show two things: First, that our algorithm converges to some point $(x^\star, y^\star)$ in a number of gradient and function evaluations which is polynomial in $1/\varepsilon, d, b, L$ (Lemma D.7, in the appendix). Secondly, roughly, we would like to show $y^\star$ is a first-order $\varepsilon$-stationary point for $f(x^\star, \cdot)$ and $x^\star$ is a $\varepsilon$-local minimum for the simulated loss function $\mathcal{L}_\varepsilon(\cdot, y^\star)$ (Lemma D.9).

**Bounding the number of gradient and function evaluations (Lemma D.7)** First, we bound the number of gradient evaluations for the maximization subroutine for $y$ (Line 6 of Algorithm 1), and then bound the number of iterations in the minimization routine for $x$ (the "While" loop in Alg. 1).

*Step 1: Bounding the number of gradient ascent steps for the maximization subroutine:* Consider the sequence of ADAM gradient ascent steps $y = y_1, y_2 \ldots, y_\ell = y'$ the max-player uses to compute her update $y'$ in Line 6 of Alg. 1. For our choice of hyperparameters, the ADAM update is $y_{j+1} = y_j + \alpha G_y(x, y_j)/(G_y^2(x, y_j))^{1/2}$, where $\alpha$ is the ADAM learning rate. Since the magnitude of our ADAM gradient satisfies $\|G_{\mathsf{y}}(x, y_j)/(G_{\mathsf{y}}^2(x, y_j))^{1/2}\|_2 = \|G_y(x_i, y_j)\|_1$, our stopping condition for gradient ascent (Line 6 of Alg. 1), which says that gradient ascent stops whenever $\|G_{\mathsf{y}}(x, y_j)\|_1 \leq \varepsilon$, implies that at each step of gradient ascent our ADAM update has magnitude at least $\alpha\varepsilon$ in $\ell_2$-norm. Using this, we then show that at each step $y_j$ of gradient ascent, we have with high probability that

$$f(y_{j+1}) - f(y_j) \geq \Omega(\alpha\varepsilon), \tag{7}$$

if the ADAM learning rate satisfies $\alpha \approx \Theta(\varepsilon/(Ld))$ (Proposition D.5). Since $f$ is $b$-bounded, this implies that the ADAM gradient ascent subroutine must terminate after $O(b/(\alpha\varepsilon)) = O(bLd/\varepsilon)$ steps.

*Step 2: Bounding the number of iterations for the minimization routine:* We first show a concentration bound for our stochastic zeroth-order oracle (Proposition D.3) and use it to show that, for our temperature schedule of $\tau_i = \tau_1/i$, after $\hat{\mathcal{I}} := \tau_1 \log(2r_{\max}b/\varepsilon^2)$ iterations, our algorithm rejects any proposed step $x' = x + \Delta$ for which $f(x', y') > f(x, y) - \varepsilon$ with probability at most roughly $1 - e^{-1/\tau_i} \leq 1 - \varepsilon^2/(2r_{\max}b)$. Therefore, roughly, with probability at least $1 - \varepsilon$, for the first $2r_{\max}b/\varepsilon$ iterations after $\hat{\mathcal{I}}$, we have that only proposed steps $x' = x + \Delta$ for which $f(x', y') \leq f(x, y) - \varepsilon$ are accepted. Moreover, our stopping condition (Line 2 in Alg. 1) stops the algorithm whenever $r_{\max}$ proposed steps are rejected in a row. Therefore, with probability at least $1 - \varepsilon$ we have that the value of the loss decreases by more than $2b/\varepsilon$ between iterations $\hat{\mathcal{I}}$ and $\hat{\mathcal{I}} + r_{\max}^2 b/\varepsilon^2$, unless our algorithm terminates during these iterations. Since $f$ is $b$-bounded, this implies our algorithm must terminate after roughly $O(r_{\max}^2 b/\varepsilon^2)$ iterations of the minimization routine (Proposition D.6).

Now, each of the $O(bLd/\varepsilon)$ steps of the maximization subroutine requires one gradient evaluation, and each of the $O(r_{\max}^2 b/\varepsilon^2)$ iterations of the minimization routine calls the maximization routine exactly once (and makes one call to stochastic oracles for the function $f$ and the $x$-gradient). Therefore, the total number of gradient and function evaluations is roughly $bLd/\varepsilon \times r_{\max}^2 b/\varepsilon^2$, which, for our choice of hyperparameter $r_{\max}$ of roughly $r_{\max} = 1/\varepsilon$, is polynomial in $1/\varepsilon, d, b, L$.

**Showing $(x^\star, y^\star)$ satisfies Inequalities equation 34 and equation 35 (Lemma D.9)**

*Step 1: Show $y^\star$ is a first-order $\varepsilon$-stationary point for $f(x^\star, \cdot)$ (equation 34)* Our stopping condition for the maximization subroutine says $\|G_y(x^\star, y^\star)\|_1 \leq \varepsilon$ at the point $y^\star$ where the subroutine terminates. To prove equation 34, we show a concentration bound for our stochastic gradient $G_y$ (the second part of Proposition D.3) and use this to show that the bound $\|G_y(x^\star, y^\star)\|_1 \leq \varepsilon$ on the stochastic gradient obtained from the stopping condition implies the desired bound on the exact gradient, $\|\overline{\nabla}_y f(x^\star, y^\star)\|_1 \leq \varepsilon$.

*Step 2: Show $x^\star$ is an $\varepsilon$-local minimum for $\mathcal{L}_\varepsilon(\cdot, y^\star)$ (equation 35)* First, we show our stopping condition for the minimization routine implies the last $r_{\max}$ steps $\Delta$ proposed by the algorithm were all rejected. This implies the last $r_{\max}$ proposed steps were sampled from the distribution $\mathcal{D}_{x^\star, y^\star}$ of the ADAM gradient $G_x(x^\star, y^\star)/(G_x^2(x^\star, y^\star))^{1/2}$ and, since they were rejected, our stopping condition implies, roughly, $f(x^\star + \Delta, y^\star) > f(x^\star, y^\star) - \varepsilon$ for all these proposed steps $\Delta$. Roughly, we use this, together with our $\mathrm{poly}(1/\varepsilon, d, b, L)$ bound on the number of iterations (Proposition D.6), to show

$$\Pr_{\Delta \sim \mathcal{D}_{x^\star, y^\star}} [f(x^\star + \Delta, y') \geq f(x^\star, y^\star) - \varepsilon] \geq 1 - \varepsilon, \tag{8}$$

where the maximization subroutine computes $y'$ by gradient ascent on $f(x^\star + \Delta, \cdot)$ initialized at $y^\star$.

To show that equation 35 holds, roughly, we would like to replace "$f$" in the bound in Ineq. equation 8 with the simulated loss function $\mathcal{L}_\varepsilon$. Towards this end, we first show that the steps traced by the gradient ascent maximization subroutine form an "$\varepsilon$-increasing" path, with endpoint $y'$, along which $f$ increases at rate at least $\varepsilon$ (Prop. D.8). Although we would ideally like to use this fact to show that $f(x^\star + \Delta, y') = \mathcal{L}_\varepsilon(x^\star + \Delta, y^\star)$, this equality does not hold in general since $\mathcal{L}_\varepsilon$ is defined using a large set of such $\varepsilon$-increasing paths, only one of which is simulated by the maximization subroutine. To get around this problem, we instead use the fact that $\mathcal{L}_\varepsilon$ is the supremum of the values of $f$ at the endpoints of *all* $\varepsilon$-increasing paths starting at $y^\star$ which seek to maximize $f(x^\star + \Delta, \cdot)$, to show that

$$f(x^\star + \Delta, y') \leq \mathcal{L}_\varepsilon(x^\star + \Delta, y^\star). \tag{9}$$

Finally, we already showed that $\|\nabla_y f(x^\star, y^\star)\|_1 \leq \varepsilon^\star$; recall from Section 2 that this implies $\mathcal{L}_\varepsilon(x^\star, y^\star) = f(x^\star, y^\star)$. Combing this with equation 8, equation 9 implies the $\varepsilon$-local minimum condition equation 35.

# C  COMPARISON OF NOTIONS OF LOCAL OPTIMALITY

In previous works Daskalakis & Panageas (2018); Heusel et al. (2017), a local saddle point (equivalently a local Nash point) has been considered as a possible notion of local min-max optimum. A point $(x^\star, y^\star)$ is said to be a local saddle point if there exists a small neighborhood around $x$ and $y$ such that $y^\star$ is a first-order stationary point for the function $f(x^\star, \cdot)$, and $x^\star$ is a local minimum for the function $f(\cdot, y^\star)$. A key difference between a local saddle point and the local min-max equilibrium we introduce in our paper (Definition 2.2) is that a local saddle point does not take into account the order in which the min-player and max-player choose $x$ and $y$.

While a local saddle point is not guaranteed to exist in general nonconvex-nonconcave min-max problems, if there exists a local saddle point, this saddle point is also a local min-max equilibrium as given by Definition 2.2. More specifically, any local saddle point for a smooth function $f$ is an $\varepsilon$-local min-max equilibrium for every $\varepsilon > 0$, provided that we pick the step size for the min-player to be small enough. This is because, if there if a point $(x^\star, y^\star)$ is a local saddle, then $y^\star$ is a first-order stationary point for the function $f(x^\star, \cdot)$. Thus, the gradient at $(x^\star, y^\star)$ must be zero and the first condition (Inequality equation 34) is satisfied.

If we pick the step size small enough, the step $\Delta$ will be such that $x^\star + \Delta$ lies within this neighborhood of $x^\star$ and hence

$$\mathcal{L}_\varepsilon(x^\star, y^\star) = f(x^\star, y^\star) \leq f(x^\star + \Delta, y^\star) \leq \mathcal{L}_\varepsilon(x^\star + \Delta, y^\star)$$

for all $\varepsilon > 0$ [6]. Thus the second condition (Inequality equation 35) is satisfied for any $\varepsilon > 0$.

Another notion of a local minimax point was proposed in Jin et al. (2020). In their definition, the max player is able to choose her move after the min-player reveals her move. The max-player is restricted to move in a small ball around $y^\star$, but is always able to find the global maximum inside this ball. In contrast, in our definition, the max-player is empowered to move much farther, as long as she is following a path along which the loss function increases rapidly. Hence, in general these two notions are incomparable. However, even under mild smoothness conditions, a local minimax point is not guaranteed to exist Jin et al. (2020), whereas a local min-max equilibrium (Definition 2.2) is.

Finally, in a parallel line of work, we prove the existence of a stronger, second-order notion of min-max equilibrium for nonconvex-nonconcave functions motivated by the notion of approximate local minimum introduced in Nesterov & Polyak (2006). The advantage of the notion in our current paper (Definition 2.2) is that it yields a concise proof of convergence and the accompanying algorithm, as our empirical results show, is effective for training GANs.

## D    MAIN RESULT

Recall that we have defined $\mathcal{D}_{x,y}$ to be the distribution of the point-wise normalized stochastic gradient $G_x(x,y)/\sqrt{G_x^2(x,y)}$. We also recall that we have made the following assumptions about our stochastic gradient, which we restate here for convenience:

**Assumption D.1** (smoothness). *Suppose that $f_t(x,y) \sim \mathcal{D}_{x,y}$ is sampled from the data distribution for any $x, y \in \mathbb{R}^d$. Then, with probability 1, $f_t$ is b-bounded, $L_1$-Lipschitz, and has L-Lipschitz gradients $\nabla f_t$.*

**Assumption D.2** (batch gradients).

$$F(x,y) = \frac{1}{\mathfrak{b}_0} \sum_{t=1}^{\mathfrak{b}_0} \mathfrak{f}_t(x,y)$$

$$G_x(x,y) = \frac{1}{\mathfrak{b}_x} \sum_{t=1}^{\mathfrak{b}_x} \nabla_x \mathfrak{f}_t(x,y),$$

$$G_y(x,y) = \frac{1}{\mathfrak{b}_y} \sum_{t=1}^{\mathfrak{b}_y} \nabla_y \mathfrak{f}_t(x,y),$$

*where, $\mathfrak{f}_1, \ldots \mathfrak{f}_t$ are sampled iid (with replacement) from the distribution $\mathcal{D}_{(x,y)}$ and $\mathfrak{b}_0, \mathfrak{b}_x, \mathfrak{b}_y > 0$ are batch sizes. Every time $G_x$, $G_y$, or $F$ is evaluated, a new, independent, batch is used.*

**Setting parameters:**    For the theoretical analysis, we set the following parameters, and we define $\frac{z}{z} := 1$ if $z = 0$. We also assume $0 < \varepsilon \le 1$.

1. $\alpha = 1$

2. $\beta_1 = 0, \beta_2 = 0, \delta = 0$

3. $\omega = \frac{\varepsilon^4}{2^{36}bLd+16}[(\tau_1 + \frac{b}{\varepsilon^2})\frac{256}{\varepsilon^2} \log(\tau_1 \frac{256}{\varepsilon^2}) \log^4(\frac{100}{\varepsilon}(\tau_1 + 1)(\frac{8b}{\varepsilon} + 1))]^{-2}$

4. $r_{\max} = \frac{128}{\varepsilon} \log^2 \left( \frac{100}{\varepsilon}(\tau_1 + 1)(8\frac{b}{\varepsilon} + 1) + \log(\frac{1}{\omega}) \right)$

5. Define $\mathcal{I} := \tau_1 \log(\frac{r_{\max}}{\omega}) + 8r_{\max}\frac{b}{\varepsilon} + 1$

6. $\eta = 2^{\frac{1}{2\mathcal{I}}} - 1$        (in particular, note that the fact that $\eta = 2^{\frac{1}{2\mathcal{I}}} - 1$ and $\mathcal{I} \ge 1$ implies that $\frac{1}{5\mathcal{I}} \le \eta \le \frac{1}{\mathcal{I}}$)

7. $\hat{\alpha} = \frac{\varepsilon(1 - \frac{1}{1+\eta})}{10Ld}$

8. Define $\mathcal{J} := \frac{20b}{3\hat{\alpha}\varepsilon}$

---

[6]Recall from Sec. 2 that if Eq. equation 34 is satisfied then $\mathcal{L}_\varepsilon(x^\star, y^\star) = f(x^\star, y^\star)$.

9. $\hat{\varepsilon}_1 = \frac{\varepsilon}{\sqrt{d}}(1 - \frac{1}{1+\eta})$

10. $\mathfrak{b}_x = 1$

11. $\mathfrak{b}_0 = \hat{\varepsilon}_1^{-2} 140^2 b^2 \log(1/\omega)$

12. $\mathfrak{b}_y = \hat{\varepsilon}_1^{-2} 140^2 L_1^2 \log(1/\omega)$

In particular, we note that $\omega \leq \frac{\varepsilon}{(32\mathcal{J}+16)\mathcal{I}}$ and $r_{\max} \geq \frac{4}{\varepsilon}\log(\frac{100\mathcal{I}}{\varepsilon})$. At every iteration $i$, where we set $\varepsilon' = \varepsilon_i$, we also have $\varepsilon' \leq \varepsilon_0(1+\eta)^{2\mathcal{I}} \leq \varepsilon$ and hence that $(\frac{\hat{\varepsilon}_1}{10} + L\sqrt{d}\hat{\alpha})\sqrt{d} \leq (1 - \frac{1}{1+\eta})\varepsilon_0 \leq (1 - \frac{1}{1+\eta})\varepsilon'$.

**Proposition D.3.** *For any $\hat{\varepsilon}_1, \omega > 0$, if we use batch sizes $\mathfrak{b}_y = \hat{\varepsilon}_1^{-2} 140^2 L_1^2 \log(1/\omega)$ and $\mathfrak{b}_0 = \hat{\varepsilon}_1^{-2} 140^2 b^2 \log(1/\omega)$, we have that*

$$\mathbb{P}\left(\|G_y(x,y) - \nabla_y f(x,y)\|_2 \geq \frac{\hat{\varepsilon}_1}{10}\right) < \omega, \tag{10}$$

*and*

$$\mathbb{P}\left(|F(x,y) - f(x,y)| \geq \frac{\hat{\varepsilon}_1}{10}\right) < \omega. \tag{11}$$

*Proof.* From Assumption D.2 we have that

$$G_y(x,y) - \nabla_y f(x,y) = \frac{1}{\mathfrak{b}_y}\sum_{t=1}^{\mathfrak{b}_y}[\nabla_y \mathsf{f}_t(x,y) - \nabla_y f(x,y)],$$

where $\mathsf{f}_1, \ldots \mathsf{f}_t$ are sampled iid (with replacement) from the data distribution $\mathfrak{D}$.

But by Assumption D.1, we have (with probability 1) that

$$\|\nabla_y \mathsf{f}_t(x,y) - \nabla_y f(x,y)\|_2 \leq \|\nabla_y \mathsf{f}_t(x,y)\|_2 + \|\nabla_y f(x,y)\|_2 \leq 2L_1.$$

Now,

$$\mathbb{E}[\nabla_y \mathsf{f}_t(x,y) - \nabla_y f(x,y)] = \mathbb{E}[\nabla_y \mathsf{f}_t(x,y) - \mathbb{E}[\nabla_y \mathsf{f}_t(x,y)]] = 0.$$

Therefore, by the Azuma-Hoefding inequality for mean-zero bounded vectors, we have

$$\mathbb{P}\left(\left\|\frac{1}{\mathfrak{b}_y}\sum_{t=1}^{\mathfrak{b}_y}[\nabla_y \mathsf{f}_t(x,y) - \nabla_y f(x,y)]\right\|_2 \geq \frac{s\sqrt{\mathfrak{b}_y}+1}{\mathfrak{b}_y}2L_1\right) < 2e^{1-\frac{1}{2}s^2} \qquad \forall s > 0.$$

Hence, if we set $s = 6\log^{1/2}(\frac{2}{\omega})$, we have that $7\log^{1/2}(\frac{2}{\omega})\sqrt{\mathfrak{b}_y}+1 \geq s\sqrt{\mathfrak{b}_y}+1$ and hence that

$$\mathbb{P}\left(\left\|\frac{1}{\mathfrak{b}_y}\sum_{t=1}^{\mathfrak{b}_y}[\nabla_y \mathsf{f}_t(x,y) - \nabla_y f(x,y)]\right\|_2 \geq \frac{7\log^{1/2}(\frac{2}{\omega})\sqrt{\mathfrak{b}_y}}{\mathfrak{b}_y}2L_1\right) < \omega.$$

Therefore,

$$\mathbb{P}\left(\left\|\frac{1}{\mathfrak{b}_y}\sum_{t=1}^{\mathfrak{b}_y}[\nabla_y \mathsf{f}_t(x,y) - \nabla_y f(x,y)]\right\|_2 \geq \frac{\hat{\varepsilon}_1}{10}\right) < \omega$$

which completes the proof of Inequality equation 10.

Inequality equation 11 follows from the exact same steps as the proof of Inequality equation 10, if we replace the bound $L_1$ for $\|\nabla_y \mathsf{f}_t(x,y)\|_2$ with the bound $b$ on $|\mathsf{f}_t(x,y)|$.

$\square$

**Proposition D.4.** *For every $j$ we have*

$$\|y_{j+1} - y_j\|_2 \leq \hat{\alpha}\sqrt{d}. \tag{12}$$

*Moreover, for every $i$ we have,*

$$\|x_{i+1} - x_i\|_2 \leq \alpha\sqrt{d}. \tag{13}$$

*Proof.*

$$\|y_{j+1} - y_j\|_2 \leq \hat{\alpha}\|m_j/\sqrt{v_j}\|_2 = \hat{\alpha}\sqrt{\sum_{k=1}^{d} \left(g_{y,j}[k]/\sqrt{g_{y,j}^2[k]}\right)^2} = \hat{\alpha}\sqrt{d}$$

This proves Inequality equation 12. The proof of Inequality equation 13 follows from the same steps as above. □

**Proposition D.5.** *Algorithm 3 terminates in at most $\mathcal{J} := \frac{20b}{3\hat{\alpha}\varepsilon}$ iterations of its "While" loop, with probability at least $1 - \omega \times \mathcal{J}$.*

*Proof.* Recalling that $/$ and $\sqrt{\cdot}$ denote element-wise operations, we have

$$\langle (m_j/\sqrt{v_j}), \ g_{y,j} \rangle = \sum_{k=1}^{d} \left(g_{y,j}[k]/\sqrt{g_{y,j}^2[k]}\right) \times g_{y,j}[k] \tag{14}$$

$$= \sum_{k=1}^{d} |g_{y,j}[k]|$$

$$= \|g_{y,j}\|_1 \geq \varepsilon' \geq \frac{\varepsilon}{2},$$

since, whenever Algorithm 3 is called by Algorithm 2, Algorithm 2 sets Algorithm 3's input $\varepsilon'$ to some value $\varepsilon' \geq \frac{\varepsilon}{2}$.

Therefore, we have that

$$\langle y_{j+1} - y_j, \ \nabla_y f(x, y_j) \rangle \overset{\text{Prop.}D.3}{\geq} \langle y_{j+1} - y_j, \ g_{y,j} \rangle - \|y_{j+1} - y_j\|_2 \times \frac{\hat{\varepsilon}_1}{10} \tag{15}$$

$$= \langle \hat{\alpha}(m_j/\sqrt{v_j}), \ g_{y,j} \rangle - \|y_{j+1} - y_j\|_2 \times \frac{\hat{\varepsilon}_1}{10}$$

$$\overset{\text{Eq.14}}{\geq} \hat{\alpha}\frac{\varepsilon}{2} - \|y_{j+1} - y_j\|_2 \times \frac{\hat{\varepsilon}_1}{10}$$

$$\overset{\text{Prop.}D.4}{\geq} \hat{\alpha}\frac{\varepsilon}{2} - \hat{\alpha}\sqrt{d} \times \frac{\hat{\varepsilon}_1}{10}$$

$$\geq \frac{4}{10}\hat{\alpha}\varepsilon,$$

with probability at least $1 - \omega$, since $\hat{\varepsilon}_1 \leq \frac{\varepsilon}{\sqrt{d}}$.

Since $f$ has $L$-Lipschitz gradient, there exits a vector $u$, with $\|u\|_2 \leq L\|y_{j+1} - y_j\|_2$, such that

$$f(y_{j+1}) - f(y_j) = \langle y_{j+1} - y_j, \ \nabla_y f(x, y_j) + u \rangle \tag{16}$$

$$= \langle y_{j+1} - y_j, \ \nabla_y f(x, y_j) \rangle + \langle y_{j+1} - y_j, \ u \rangle$$

$$\overset{\text{Eq.15}}{\geq} \frac{4}{10}\hat{\alpha}\varepsilon - L\|y_{j+1} - y_j\|_2^2$$

$$\overset{\text{Prop.}D.4}{\geq} \frac{4}{10}\hat{\alpha}\varepsilon - L\hat{\alpha}^2 d$$

$$\geq \frac{3}{10}\hat{\alpha}\varepsilon,$$

with probability at least $1 - \omega$, since $\hat{\alpha} \leq \frac{\varepsilon}{10Ld}$.

Since $f$ takes values in $[-b, b]$, Inequality equation 16 implies that Algorithm 3 terminates in at most $\mathcal{J} = \frac{20b}{3\hat{\alpha}\varepsilon}$ iterations, with probability at least $1 - \omega \times \mathcal{J}$.

$\square$

**Proposition D.6.** *Algorithm 2 terminates in at most $\mathcal{I} := \tau_1 \log(\frac{r_{\max}}{\omega}) + 8r_{\max}\frac{b}{\varepsilon} + 1$ iterations of its "While" loop, with probability at least $1 - 2\omega \times (r_{\max}\frac{2b}{\frac{1}{4}\varepsilon} + 1)$.*

*Proof.* For any $i > 0$, let $E_i$ be the "bad" event that both $f(x_{i+1}, y_{i+1}) - f(x_i, y_i) > -\frac{\varepsilon}{4}$ and $\mathsf{Accept}_i = \mathsf{True}$.

Then by Proposition D.3, since $\frac{\hat{\varepsilon}_1}{10} \leq \frac{\varepsilon}{8}$, we have that

$$\mathbb{P}(E_i) \leq e^{-\frac{i}{\tau_1}} + \omega. \tag{17}$$

Define $\hat{\mathcal{I}} := \tau_1 \log(\frac{r_{\max}}{\omega})$.

Then for $i \geq \hat{\mathcal{I}}$, from Line 12 of Algorithm 2 we have by Inequality equation 17 that

$$\mathbb{P}(E_i) \leq 2\omega.$$

Define $h := r_{\max}\frac{2b}{\frac{1}{4}\varepsilon} + 1$. Then

$$\mathbb{P}\left(\bigcup_{i=\hat{\mathcal{I}}}^{\hat{\mathcal{I}}+h} E_i\right) \leq 2\omega \times h. \tag{18}$$

Since $f$ takes values in $[-b, b]$, if $\bigcup_{i=\hat{\mathcal{I}}}^{\hat{\mathcal{I}}+h} E_i$ does not occur, the number of accepted steps over the iterations $\hat{\mathcal{I}} \leq i \leq \hat{\mathcal{I}} + h$ (that is, the size of the set $\{i : \hat{\mathcal{I}} \leq i \leq \hat{\mathcal{I}} + h, \mathsf{Accept}_i = \mathsf{True}\}$) is at most $\frac{2b}{\frac{1}{4}\varepsilon}$.

Therefore, since $h = r_{\max}\frac{2b}{\frac{1}{4}\varepsilon} + 1$, we must have that there exists a number $\mathsf{i}$, with $\hat{\mathcal{I}} \leq \mathsf{i} \leq \mathsf{i} + r_{\max} \leq \hat{\mathcal{I}} + h$, such that $\mathsf{Accept}_i = \mathsf{False}$ for all $i \in [\mathsf{i}, \mathsf{i} + r_{\max}]$.

Therefore the condition in the While loop (Line 9) of Algorithm 2 implies that Algorithm 2 terminates after at most $\mathsf{i} + r_{\max} \leq \hat{\mathcal{I}} + h$ iterations of its While loop, as long as $\bigcup_{i=\hat{\mathcal{I}}}^{\hat{\mathcal{I}}+h} E_i$ does not occur.

Therefore, Inequality 18 implies that, with probability at least $1 - 2\omega \times (r_{\max}\frac{2b}{\frac{1}{4}\varepsilon} + 1)$, Algorithm 2 terminates after at most

$$\hat{\mathcal{I}} + h = \tau_1 \log(\frac{r_{\max}}{\omega}) + 8r_{\max}\frac{b}{\varepsilon} + 1$$

iterations of its "While" loop.

$\square$

**Lemma D.7.** *Algorithm 2 terminates after at most $(\tau_1 \log(\frac{r_{\max}}{\omega}) + 4r_{\max}\frac{b}{\varepsilon} + 1) \times (\mathcal{J} \times \mathfrak{b}_y + \mathfrak{b}_0 + \mathfrak{b}_x)$ gradient and function evaluations.*

*Proof.* Each iteration of the While loop in Algorithm 2 computes one batch gradient with batch size $\mathfrak{b}_x$, and one stochastic function evaluation of batch size $\mathfrak{b}_0$, and calls Algorithm 3 exactly once.

Each iteration of the While loop in Algorithm 3 computes one batch gradient with batch size $\mathfrak{b}_y$.

The result then follows directly from Propositions D.6 and D.5.

$\square$

Recall the paths $\gamma(t)$ from Definition 2.1. From now on we will refer to such paths as "$\varepsilon$-increasing paths". That is, for any $\varepsilon > 0$, we say that a path $\gamma(t)$ is an "$\varepsilon$-increasing path" if at every point along this path we have that $\left\| \frac{\mathrm{d}}{\mathrm{d}t} \gamma(t) \right\|_\infty$ and that $\frac{\mathrm{d}}{\mathrm{d}t} f(x, \gamma(t)) > \varepsilon$ (Inequality equation 33).

**Proposition D.8.** *Every time Algorithm 3 is called we have that, with probability at least $1 - 2\omega\mathcal{J}$, the path consisting of the line segments $[y_j, y_{j+1}]$ formed by the points $y_j$ computed by Algorithm 3 has a parametrization $\gamma(t)$ which is an $\frac{1}{1+\eta}\varepsilon'$-increasing path.*

*Proof.* We consider the following continuous parametrized path $\gamma(t)$:

$$\gamma(t) = y_j + tv_j \qquad t \in [\hat{\alpha}(j-1), \hat{\alpha}j], \qquad j \in [j_{\max}],$$

where $v_j := g_{y,j}/\sqrt{(g_{y,j})^2}$ and $j_{\max}$ is the number of iterations of the While loop of Algorithm 3.

First, we show that this path has at most at most unit-velocity in the infinity-norm at all but a finite number of time-points. For any point in one of the open intervals $t \in (\hat{\alpha}(j-1), \hat{\alpha}j)$, we have that [7]

$$\left\| \frac{\mathrm{d}}{\mathrm{d}t} \gamma(t) \right\|_\infty = \|v_j\|_\infty = \|g_{y,j}/\sqrt{(g_{y,j})^2}\|_\infty \leq \|(1, \ldots, 1)\|_\infty = 1.$$

This implies that the path $\gamma(t)$ has unit velocity in the infinity norm at all but a finite number of time-points.

Next, we show that $\frac{\mathrm{d}}{\mathrm{d}t} f(x, \gamma(t)) \geq \varepsilon'$.

Now,

$$\|v_j\|_2 = \sqrt{d} \tag{19}$$

and hence,

$$\|y_{j+1} - y_j\|_2 = \|\hat{\alpha}v_j\|_2 = \hat{\alpha}\sqrt{d} \tag{20}$$

at every step $j$.

By Line 12 of Algorithm 3 we have that $\|g_{y,j}/\sqrt{|g_{y,j}|}\|_2^2 > \varepsilon'$ for every $j \in [j_{\max}]$, where we define $j_{\max} \in \mathbb{N} \cup \{\infty\}$ to be the number of iterations of the While loop in Algorithm 3. Therefore, for every $j \in [j_{\max}]$, by Proposition D.3 we have with probability at least $1 - \omega$ that

$$\frac{\mathrm{d}}{\mathrm{d}t} f(x, \gamma(t)) \geq [\nabla_y f(x, y_j) - L\|y_{j+1} - y_j\|_2 u]^\top v_j \tag{21}$$

$$\overset{\text{Eq.20}}{=} [\nabla_y f(x, y_j) - L\sqrt{d}\hat{\alpha}u]^\top v_j$$

$$= [g_{y,j} - \frac{\hat{\varepsilon}_1}{10}w - L\sqrt{d}\hat{\alpha}u]^\top v_j$$

$$= g_{y,j}^\top v_j - [\frac{\hat{\varepsilon}_1}{10}w + L\sqrt{d}\hat{\alpha}u]^\top v_j$$

$$= g_{y,j}^\top \left( g_{y,j}/\sqrt{(g_{y,j})^2} \right) - [\frac{\hat{\varepsilon}_1}{10}w + L\sqrt{d}\hat{\alpha}u]^\top v_j$$

$$= \|g_{y,j}/\sqrt{|g_{y,j}|}\|_2^2 - [\frac{\hat{\varepsilon}_1}{10}w + L\sqrt{d}\hat{\alpha}u]^\top v_j$$

$$\geq \|g_{y,j}/\sqrt{|g_{y,j}|}\|_2^2 - (\frac{\hat{\varepsilon}_1}{10} + L\sqrt{d}\hat{\alpha})\|v_j\|_2$$

$$\overset{\text{Eq.19}}{=} \|g_{y,j}/\sqrt{|g_{y,j}|}\|_2^2 - (\frac{\hat{\varepsilon}_1}{10} + L\sqrt{d}\hat{\alpha})\sqrt{d}$$

$$\geq \varepsilon' - (\frac{\hat{\varepsilon}_1}{10} + L\sqrt{d}\hat{\alpha})\sqrt{d}$$

$$\geq \frac{1}{1+\eta}\varepsilon' \qquad \forall t \in [\hat{\alpha}(j-1), \hat{\alpha}j],$$

---

[7] Recall that we use the convention $0/0 = 0$ when computing the Adam update in our algorithm. This implies that $|g_{y,j}/\sqrt{(g_{y,j})^2}| \leq (1, \ldots, 1)$.

for some unit vectors $u, w \in \mathbb{R}^d$, since $(\frac{\hat{\varepsilon}_1}{10} + L\sqrt{d}\hat{\alpha})\sqrt{d} \leq (1 - \frac{1}{1+\eta})\varepsilon_0 \leq (1 - \frac{1}{1+\eta})\varepsilon'$.

But by Proposition D.5 we have that $j_{\max} \leq \mathcal{J}$ with probability at least $1 - \omega \times \mathcal{J}$. Therefore inequality equation 21 implies that

$$\frac{\mathrm{d}}{\mathrm{d}t} f(\mathsf{x}, \gamma(t)) \geq \frac{1}{1+\eta}\varepsilon' \qquad \forall t \in [\hat{\alpha}(j-1), \hat{\alpha}j] \qquad \forall j \in [j_{\max}],$$

with probability at least $1 - 2\omega\mathcal{J}$.

$\square$

**Lemma D.9.** *Let $i^\star$ be such that $i^\star - 1$ is the last iteration $i$ of the "While" loop in Algorithm 2 for which $\mathsf{Accept}_i = \mathsf{True}$. Then with probability at least $1 - 2\omega\mathcal{J}\mathcal{I} - 2\omega \times (r_{\max}\frac{2b}{\frac{1}{4}\varepsilon} + 1)$ we have that*

$$\|\nabla_y f(x^\star, y^\star)\|_1 \leq \frac{1}{\sqrt{1+\eta}}\varepsilon_{i^\star}. \tag{22}$$

*Moreover, with probability at least $1 - \frac{\varepsilon}{100} - 2\omega \times (r_{\max}\frac{2b}{\frac{1}{4}\varepsilon} + 1)$ we have that*

$$\mathbb{P}_{\Delta \sim \mathcal{D}_{x^\star, y^\star}}\left(\mathcal{L}_{\varepsilon_{i^\star}}(x^\star + \Delta, y^\star) \leq \mathcal{L}_{\varepsilon_{i^\star}}(x^\star, y^\star) - \frac{1}{2}\varepsilon \,\Big|\, x^\star, y^\star\right) \leq \frac{1}{2}\varepsilon. \tag{23}$$

*and that*

$$\frac{\varepsilon}{2} \leq \varepsilon_{i^\star} \leq \varepsilon. \tag{24}$$

*Proof.* First, we note that $(x^\star, y^\star) = (x_i, y_i)$ for all $i \in \{i^\star, \ldots, i^\star + r_{\max}\}$, and that Algorithm 2 stops after exactly $i^\star + r_{\max}$ iterations of the "While" loop in Algorithm 2.

Define $\Delta_i := g_{\mathsf{x},i}/\sqrt{(g_{\mathsf{x},i})^2}$ for every $i$. Then $\Delta_i \sim \mathcal{D}_{x_i, y_i}$.

Let $\mathsf{H}_i$ be the "bad" event that, when Algorithm 3 is called during the $i$th iteration of the "While" loop in Algorithm 2, the path traced by Algorithm 3 is not an $\varepsilon_i$-increasing path. Then, by Proposition D.8 we have that

$$\mathbb{P}(\mathsf{H}_i) \leq 2\omega\mathcal{J}. \tag{25}$$

Let $\mathsf{K}_i$ be the "bad" event that $\|G_y(x_i, y_i) - \nabla_y f(x_i, y_i)\|_2 \geq \frac{\hat{\varepsilon}_1}{10}$. Then by Propositions D.3 and D.5 we have that

$$\mathbb{P}(\mathsf{K}_i) \leq 2\omega\mathcal{J}. \tag{26}$$

Whenever $\mathsf{K}_i^c$ occurs we have that

$$\begin{aligned}
\|\nabla_y f(x_i, y_i)\|_1 &\leq \|G_y(x_i, y_i)\|_1 + \|G_y(x_i, y_i) - \nabla_y f(x_i, y_i)\|_1 \qquad (27)\\
&\leq \varepsilon_i + \sqrt{d}\|G_y(x_i, y_i) - \nabla_y f(x_i, y_i)\|_2\\
&\leq \frac{1}{1+\eta}\varepsilon_i + \frac{\hat{\varepsilon}_1}{10}\\
&\leq \frac{1}{\sqrt{1+\eta}}\varepsilon_i,
\end{aligned}$$

where the second Inequality holds by Line 12 of Algorithm 3, and the last inequality holds since $\frac{\hat{\varepsilon}_1}{10} \leq 1 - \frac{1}{\sqrt{1+\eta}}$.

Therefore, Inequalities equation 26 and equation 27 together with Proposition D.6 imply that

$$\|\nabla_y f(x^\star, y^\star)\|_1 \leq \frac{1}{\sqrt{1+\eta}}\varepsilon_{i^\star}$$

with probability at least $1 - 2\omega\mathcal{J}\mathcal{I} - 2\omega \times (r_{\max}\frac{2b}{\frac{1}{4}\varepsilon} + 1)$. This proves Inequality equation 22.

Inequality equation 27 also implies that, whenever $\mathsf{K}_i^c$ occurs, the set $S_{\varepsilon_i, x_i, y_i}$ of $\varepsilon_i$-increasing paths with initial point $y_i$ (and $x$-value $x_i$) consists only of the single point $y_i$. Therefore, we have that

$$\mathcal{L}_{\varepsilon_i}(x_i, y_i) = f(x_i, y_i) \tag{28}$$

whenever $\mathsf{K}_i^c$ occurs.

Moreover, whenver $\mathsf{H}_i^c$ occurs we have that $\mathcal{Y}_{i+1}$ is the endpoint of an $\varepsilon_i$-increasing path with starting point $(x_i + \Delta_i, y_i)$. Now, $\mathcal{L}_{\varepsilon_i}(x_i + \Delta_i, y_i)$ is the supremum of the value of $f$ at the endpoints of all $\varepsilon_i$-increasing paths with starting point $(x_i + \Delta_i, y_i)$. Therefore, we must have that

$$\mathcal{L}_{\varepsilon_i}(x_i + \Delta_i, y_i) \geq f(x_i + \Delta_i, \mathcal{Y}_{i+1}) \tag{29}$$

whenever $\mathsf{H}_i^c$ occurs.

Therefore,

$$\mathbb{P}_{\Delta \sim \mathcal{D}_{x_i, y_i}} \left( \mathcal{L}_{\varepsilon_i}(x_i + \Delta, y_i) > \mathcal{L}_{\varepsilon_i}(x_i, y_i) - \frac{1}{2}\varepsilon \Big| x_i, y_i \right) \tag{30}$$

$$\overset{\text{Eq.28,29}}{\geq} \mathbb{P}_{\Delta \sim \mathcal{D}_{x_i, y_i}} \left( f(x_i + \Delta_i, \mathcal{Y}_{i+1}) > f(x_i, y_i) - \frac{1}{2}\varepsilon \Big| x_i, y_i \right) - \mathbb{P}(\mathsf{H}_i) - \mathbb{P}(\mathsf{K}_i)$$

$$\overset{\text{Prop.}D.3}{\geq} \mathbb{P}_{\Delta \sim \mathcal{D}_{x_i, y_i}} \left( F(x_i + \Delta, \mathcal{Y}_{i+1}) > F(x_i, y_i) - \frac{1}{4}\varepsilon \Big| x_i, y_i \right) - 2\omega - \mathbb{P}(\mathsf{H}_i) - \mathbb{P}(\mathsf{K}_i)$$

$$\geq \mathbb{P}\left( \mathsf{Accept}_i = \mathsf{False} \big| x_i, y_i \right) - 2\omega - \mathbb{P}(\mathsf{H}_i) - \mathbb{P}(\mathsf{K}_i)$$

$$\overset{\text{Eq.25,26}}{\geq} \mathbb{P}\left( \mathsf{Accept}_i = \mathsf{False} \big| x_i, y_i \right) - 2\omega - 2\omega\mathcal{J} - 2\omega\mathcal{J}, \qquad \forall i \leq \mathcal{I},$$

where the second inequality holds by Proposition D.3, since $\frac{\hat{\varepsilon}_1}{10} \leq \frac{\varepsilon}{8}$.

Define $p_i := \mathbb{P}_{\Delta \sim \mathcal{D}_{x_i, y_i}} \left( \mathcal{L}_{\varepsilon_i}(x_i + \Delta, y_i) > \mathcal{L}_{\varepsilon_i}(x_i, y_i) - \frac{1}{2}\varepsilon \Big| x_i, y_i \right)$ for every $i \in \mathbb{N}$. Then Inequality equation 30 implies that

$$\mathbb{P}\left( \mathsf{Accept}_i = \mathsf{False} \big| x_i, y_i \right) \leq p_i + \omega(4\mathcal{J} + 2) \leq p_i + \frac{1}{8}\varepsilon \qquad \forall i \leq \mathcal{I}, \tag{31}$$

since $\omega \leq \frac{\varepsilon}{32\mathcal{J}+16}$.

We now consider what happens for indices $i$ for which $p_i \leq 1 - \frac{1}{2}\varepsilon$. Since $(x_{i+s}, y_{i+s}) = (x_i, y_i)$ whenever $\mathsf{Accept}_{i+k} = \mathsf{False}$ for all $0 \leq k \leq s$, we have by Inequality equation 31 that

$$\mathbb{P}\left( \cap_{s=0}^{r_{\max}} \{\mathsf{Accept}_{i+s} = \mathsf{False}\} \Big| p_i \leq 1 - \frac{1}{2}\varepsilon \right) \leq (1 - \frac{1}{4}\varepsilon)^{r_{\max}} \leq \frac{\varepsilon}{100\mathcal{I}} \qquad \forall i \leq \mathcal{I} - r_{\max}$$

since $r_{\max} \geq \frac{4}{\varepsilon} \log(\frac{100\mathcal{I}}{\varepsilon})$.

Therefore, with probability at least $1 - \frac{\varepsilon}{100\mathcal{I}} \times \mathcal{I} = 1 - \frac{\varepsilon}{100}$, we have that the event $\cap_{s=0}^{r_{\max}} \{\mathsf{Accept}_{i+s} = \mathsf{False}\}$ does not occur for any $i \leq \mathcal{I} - r_{\max}$ for which $p_i \leq 1 - \frac{1}{2}\varepsilon$.

Recall from Proposition D.6 that Algorithm 2 terminates in at most $\mathcal{I}$ iterations of its "While" loop, with probability at least $1 - 2\omega \times (r_{\max \frac{1}{4}\varepsilon} + 1)$.

Therefore,

$$\mathbb{P}\left( p_{i^\star} > 1 - \frac{1}{2}\varepsilon \right) \geq 1 - \frac{\varepsilon}{100} - 2\omega \times (r_{\max \frac{1}{4}\varepsilon} + 1). \tag{32}$$

In other words, by the definition of $p_{i^\star}$, Inequality equation 32 implies that with probability at least $1 - \frac{\varepsilon}{100} - 2\omega \times (r_{\max \frac{1}{4}\varepsilon} + 1)$, the point $(x^\star, y^\star)$ is such that

$$\mathbb{P}_{\Delta \sim \mathcal{D}_{x^\star, y^\star}} \left( \mathcal{L}_{\varepsilon_{i^\star}}(x^\star + \Delta, y^\star) \leq \mathcal{L}_{\varepsilon_{i^\star}}(x^\star, y^\star) - \frac{1}{2}\varepsilon \Big| x^\star, y^\star \right) \leq \frac{1}{2}\varepsilon.$$

This completes the proof of inequality equation 23.

Finally we note that when Algorithm 2 terminates in at most $\mathcal{I}$ iterations of its "While" loop, we have $\varepsilon_{i^\star} = \varepsilon_0(1+\eta)^{2i^\star} \leq \varepsilon_0(1+\eta)^{2\mathcal{I}} \leq \varepsilon$. This completes the proof on Inequality equation 24.   $\square$

We can now complete the proof of the main theorem:

*Proof of Theorem 2.3.* First, by Lemma D.7 our algorithm converges to some point $(x^\star, y^\star)$ after at most $(\tau_1 \log(\frac{r_{\max}}{\omega}) + 4r_{\max}\frac{b}{\varepsilon} + 1) \times (\mathcal{J} \times \mathfrak{b}_y + \mathfrak{b}_0 + \mathfrak{b}_x)$ gradient and function evaluations, which is polynomial in $1/\varepsilon, d, b, L_1, L$. In particular, for $L, b \geq 1$, the number of gradient and function evaluations is $\tilde{O}(d^2 L^2 b^6 \varepsilon^{-11})$.

By Lemma D.9, if we set $\varepsilon^\star = \varepsilon_{i^\star}$, we have that Inequalities equation 34 and equation 35 hold for $\varepsilon^\star \in [\frac{1}{2}\varepsilon, \varepsilon]$ with probability at least $1 - 2\omega\mathcal{J}\mathcal{I} - 2\omega \times (r_{\max}\frac{2b}{\frac{1}{4}\varepsilon} + 1) \geq \frac{9}{10}$.   $\square$

# E   SIMULATION SETUP

In this section we discuss the neural network architectures, choice of hyperparameters, and hardware used for both the real and synthetic data simulations.

**Datasets.** We evaluate the performance of our algorithm on both real-world and synthetic data. The real-world datasets used are MNIST and CIFAR-10. In one of our MNIST simulations we train our algorithm on the entire MNIST dataset, and on the remaining simulations we train on the subset of the MNIST dataset which includes only the digits labeled 0 or 1 (note however that the algorithms do not see the labels). For simplicity, we refer to this dataset as the 0-1 MNIST dataset.

The synthetic dataset we use consists of 512 points points sampled at the start of each simulation from a mixture of four equally weighted Gaussians in two dimensions with standard deviation 0.01; and means positioned at $(0, 1)$, $(1, 0)$, $(-1, 0)$ and $(0, -1)$.

For all of our simulations on both real and synthetic datasets, following Goodfellow et al. (2014) and Metz et al. (2017), we use the cross entropy loss function for training, where $f(x, y) = \log(D_y(\zeta)) + \log(1 - D_y(G_x(\xi)))$, where $x$ are the weights of the generator's neural network and $y$ are the weights of the discriminator's neural network, where $\zeta$ is sampled from the data distribution, and $\xi$ is a Gaussian with identity covariance matrix. The neural network architectures and hyperparameters for both the real and synthetic data simulations are specified in the following paragraphs.

**Hyperparameters for MNIST simulations.** For the MNIST simulations, we use a batch size of 128, with Adam learning rate of 0.0002 and hyperparameter $\beta_1 = 0.5$ for both the generator and discriminator gradients. Our code for the MNIST simulations is based on the code of Renu Khandelwal Khandelwal (2019) and Rowel Atienza Atienza (2017), which originally used gradient descent ascent and ADAM gradients for training.

For the generator we use a neural network with input of size 256 and 3 hidden layers, with leaky RELUS each with "alpha" parameter 0.2 and dropout regularization of 0.2 at each layer. The first layer has size 256, the second layer has size 512, and the third layer has size 1024, followed by an output layer with hyperbolic tangent ("tanh") acvtivation.

For the discriminator we use a neural network with 3 hidden layers, and leaky RELUS each with "alpha" parameter 0.2, and dropout regularization of 0.3 (for the first two layers) and 0.2 (for the last layer). The first layer has size 1024, the second layer has size 512, the third layer has size 256, and the hidden layers are followed by a projection to 1 dimension with sigmoid activation (which is fed as input to the cross entropy loss function).

**Hyperparameters for Gaussian mixture simulations.** For the simulations on Gaussian mixture data, we have used the code provided by the authors of Metz et al. (2017), which uses a batch size 512, Adam learning rates of $10^{-3}$ for the generator and $10^{-4}$ for the discriminator, and Adam parameter $\beta_1 = 0.5$ for both the generator and discriminator.[8]

---

[8]Note that the authors also mention using slightly different ADAM parameters and neural network architecture in their paper than in their code; we have used the Adam parameters and neural network architecture provided in their code.

We use the same neural networks that were used in the code from Metz et al. (2017): The generator uses a fully connected neural network with 2 hidden layers of size 128 and RELU activation, followed by a linear projection to two dimensions. The discriminator uses a fully connected neural network with 2 hidden layers of size 128 and RELU activation, followed by a linear projection to 1 dimension (which is fed as input to the cross entropy loss function). As in the paper Metz et al. (2017), we initialize all the neural network weights to be orthogonal with scaling 0.8.

**Hyperparameters for CIFAR-10 simulations.** For the CIFAR-10 simulations, we use a batch size of 128, with Adam learning rate of 0.02 and hyperparameter $\beta_1 = 0.5$ for both the generator and discriminator gradients. Our code for the CIFAR-10 simulations is based on the code of Jason Brownlee Brownlee (2019), which originally used gradient descent ascent and ADAM gradients for training.

For the generator we use a neural network with input of size 100 and 4 hidden layers. The first hidden layer consists of a dense layer with $4,096$ parameters, followed by a leaky RELU layer, whose activations are reshaped into $246 \ 4 \times 4$ feature maps. The feature maps are then upscaled to an output shape of 32 x 32 via three hidden layers of size 128 each consisting of a convolutional *Conv2DTranspose* layer followed by a leaky RELU layer, until the output layer where three filter maps (channels) are created. Each leaky RELU layer has "alpha" parameter 0.2.

For the discriminator, we use a neural network with input of size $32 \times 32 \times 3$ followed by 5 hidden layers. The first four hidden layers each consist of a convolutional *Conv2DTranspose* layer followed by a leaky RELU layer with "alpha" parameter 0.2. The first layer has size 64, the next two layers each have size 128, and the fourth layer has size 256. The output layer consists of a projection to 1 dimension with dropout regularization of 0.4 and sigmoid activation function.

FID scores were computed every 2500 iterations. To compute each FID score we used 10,000 randomly selected images from the CIFAR-10 dataset, and 10,000 generated images.

**Setting hyperparameters.** In our simulations, our goal was to be able to use the smallest number of discriminator or unrolled steps while still learning the distribution in a short amount of time, and we therefore decided to compare all algorithms using the same hyperparameter $k$. To choose this single value of $k$, we started by running each algorithm with $k = 1$ and increased the number of discriminator steps until one of the algorithms was able to learn the distribution consistently in the first 1500 iterations. This resulted in a choice of $k = 1$ for the MNIST datasets and a choice of $k = 6$ for the Gaussian mixture model data. For the CIFAR-10 dataset we simply used $k = 1$ for both algorithms (since for CIFAR-10 it is difficult to visually determine if all modes were learned).

Our temperature hyper-parameter was set by running our algorithm with temperatures in the set $\{1, 2, 3, 4, 5, 10\}$, and choosing the temperature which gave the best performance.

**Hardware.** Our simulations on the MNIST, 0-1 MNIST, and Gaussian datasets were performed on four 3.0 GHz Intel Scalable CPU Processors, provided by AWS.

Our simulations on the CIFAR-10 dataset were performed on one GPU with High frequency Intel Xeon E5-2686 v4 (Broadwell) processors, provided by AWS.

## F   ADDITIONAL SIMULATION RESULTS

In this section we show additional results from simulations which we did not have space to include in the main body of the paper.

### F.1   OUR ALGORITHM ON THE FULL MNIST DATASET

Here we show the results of the simulation of our algorithm on the full MNIST dataset (Fig. 4).

### F.2   OUR ALGORITHM AND GDA TRAINED ON THE CIFAR-10 DATASET.

In this section we show the results of all the runs of the simulations of our algorithm and GDA on the CIFAR-10 dataset, which were mentioned in Section 3.1 (Figures 5 and 6).

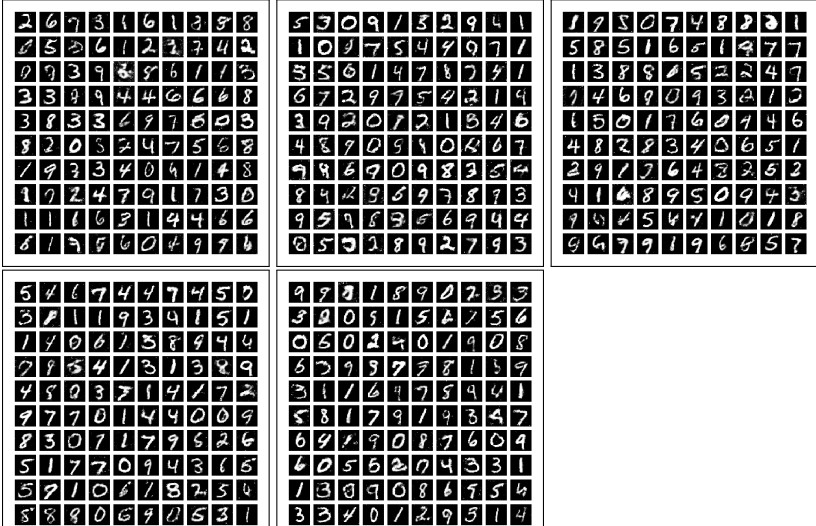

Figure 4: We ran our algorithm (with $k = 1$ discriminator steps and acceptance rate $e^{-\frac{1}{\tau}} = \frac{1}{5}$) on the full MNIST dataset for 39,000 iterations, and then plotted images generated from the resulting generator. We repeated this simulation five times; the generated images from each of the five runs are shown here.

### F.3    OUR ALGORITHM TRAINED ON THE 0-1 MNIST DATASET.

Here we show the results of the simulations of our algorithm on 0-1 MNIST which were mentioned in Section 3.1 (Figures 7 and 8).

### F.4    COMPARISON WITH GDA ON MNIST

In this section we show the results from the different runs of the simulations of our algorithm and GDA on the 0-1 MNIST dataset, which were mentioned in Section 3.2 (Figures 9 and 10).

### F.5    RANDOMIZED ACCEPTANCE RULE WITH DECREASING TEMPERATURE

In this section we give the simulations mentioned in the paragraph towards the beginning of Section 3, which discusses simplifications to our algorithm. We included these simulations to check whether our algorithm also works well when it is implemented using a randomized acceptance rule with a decreasing temperature schedule.

### F.6    COMPARISON OF ALGORITHMS ON MIXTURE OF 4 GAUSSIANS

In this section we show the results of all the runs of the simulation mentioned in Figure 2, where all the algorithms were trained on a 4-Gaussian mixture dataset for 1500 iterations. For each run, we plot points from the generated distribution at iteration 1,500. Figure 13 gives the results for GDA with $k = 1$ discriminator step. Figure 14 gives the results for GDA with $k = 6$ discriminator steps. Figure 15 gives the results for the Unrolled GANs algorithm. Figure 16 gives the results for our algorithm.

## GDA

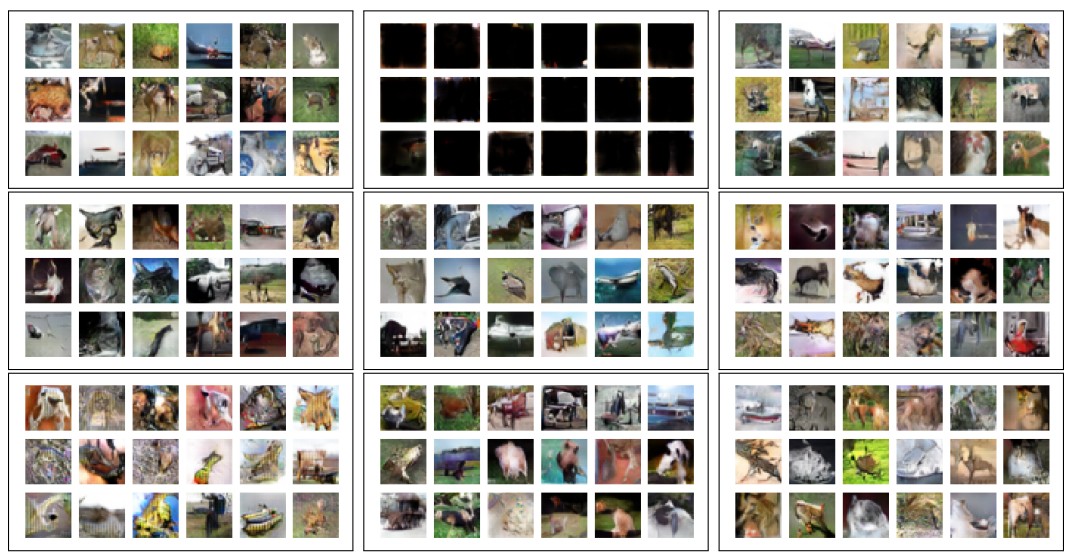

## Our Algorithm

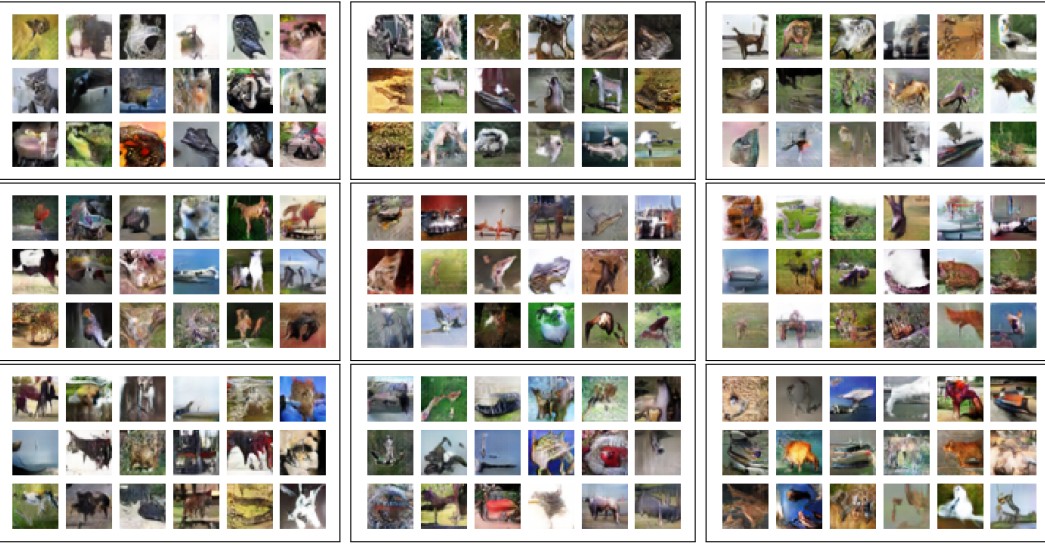

Figure 5: GAN trained using our algorithm (with $k = 1$ discriminator steps and acceptance rate $e^{-1/\tau} = 1/2$) and GDA. We repeated this simulation nine times; we display here images generated from the resulting generator for each of the nine runs of GDA (top) and our algorithm (bottom). The final FID scores at 50,000 iterations for each of the nine runs (corresponding to the images above from left to right and then top to bottom) were {35.6, 36.3, 33.8, 35.2, 34.5, 36.7, 34.9, 36.9, 36.6} for our algorithm and {33.0, 197.1, 34.3, 34.3, 33.8, 37.0, 45.3, 34.7, 34.7} for GDA.

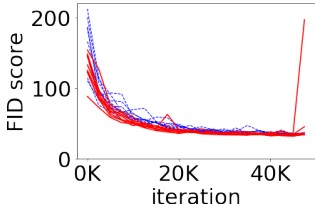

Figure 6: Plots of FID scores for GANs trained using our algorithm (with $k = 1$ discriminator steps and acceptance rate $e^{-1/\tau} = {}^{1}/{}_{2}$) and GDA on CIFAR-10 for 50,000 iterations. We repeated this simulation nine times, and plotted the FID scores for our algorithm (dashed blue) and GDA (solid red).

**Our algorithm**

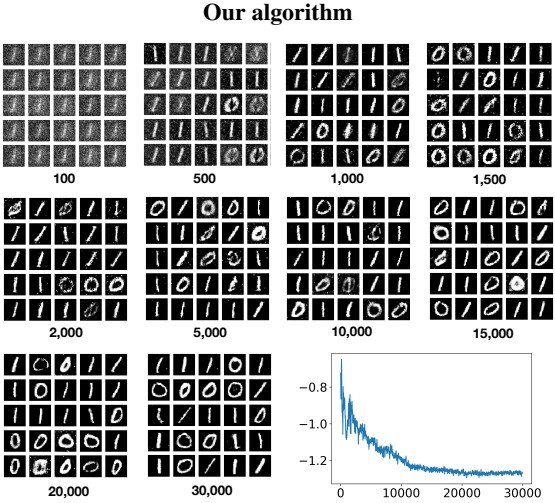

Figure 7: We trained our algorithm on the 0-1 MNIST dataset for 30,000 iterations (with $k = 1$ discriminator steps and acceptance rate $e^{-\frac{1}{\tau}} = \frac{1}{5}$). We repeated this experiment five times. For one of the runs, we plotted 25 generated images produced by the generator at various iterations. We also plotted a moving average of the computed loss function values, averaged over a window size of 50.

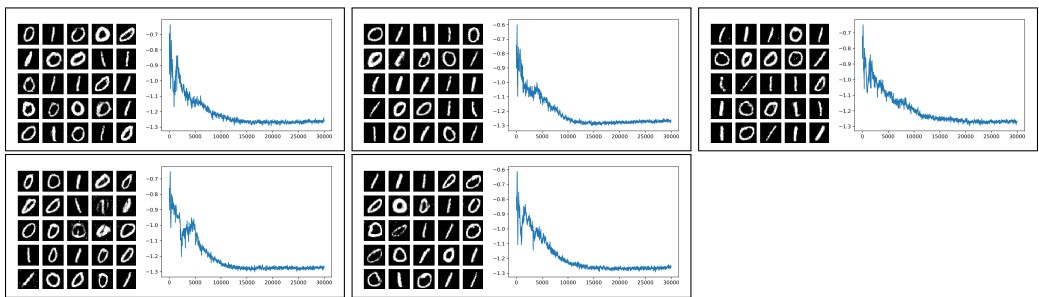

Figure 8: We show the images generated at the 30,000'th iteration for all 5 runs of the simulation in Figure 7, together with a plot of their computed loss function values. Recall from the caption of Figure 7 that we trained our algorithm on the 0-1 MNIST dataset for 30,000 iterations (with $k = 1$ discriminator steps and acceptance rate $e^{-\frac{1}{\tau}} = \frac{1}{5}$). We repeated this experiment five times. For one of the runs, we plotted 25 generated images produced by the generator at various iterations. We also plotted a moving average of the computed loss function values, averaged over a window size of 50.

Figure 9: Images generated at the 1000'th iteration of the 13 runs of the GDA simulation mentioned in Figure 1. In 77% of the runs the generator seems to be generating only 1's at the 1000'th iteration.

Figure 10: Images generated at the 1000'th iteration of each of the 22 runs of our algorithm for the simulation mentioned in in Figure 1.

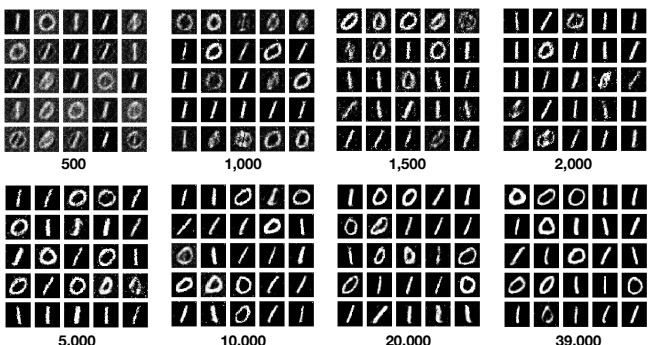

Figure 11: In this simulation we used a randomized accept/reject rule, with a decreasing temperature schedule. The algorithm was run for 39,000 iterations, with a temperature schedule of $e^{-\frac{1}{\tau_i}} = \frac{1}{4+e^{(i/20000)^2}}$. Proposed steps which decreased the computed value of the loss function were accepted with probability 1, and proposed steps which increased the computed value of the loss function were rejected with probability $\max(0, 1 - e^{-\frac{i}{\tau_1}})$ at each iteration $i$. We ran the simulation 5 times, and obtained similar results each time, with the generator learning both modes. In this figure, we plotted the generated images from one of the runs at various iterations, with the iteration number specified at the bottom of each figure (see also Figure 12 for results from the other four runs)

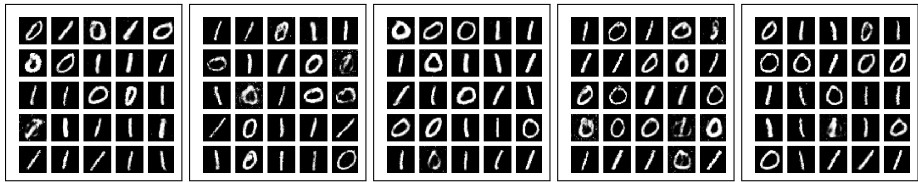

Figure 12: Images generated at the 39,000'th iteration of each of the 5 runs of our algorithm for the simulation mentioned in Figure 11 with a randomized acceptance rule with a temperature schedule of $e^{-\frac{1}{\tau_i}} = \frac{1}{4 + e^{(i/20000)^2}}$.

**GDA**

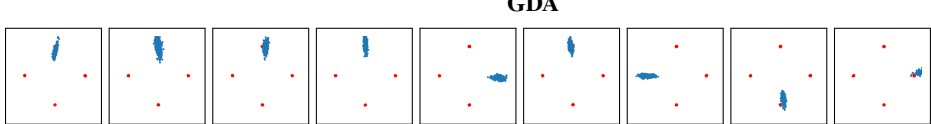

Figure 13: The generated points at the 1500'th iteration for all 9 runs of the GDA algorithm with $k = 1$ discriminator step, for the simulation mentioned in Figure 2. At the 1500'th iteration, GDA had learned exactly one mode for each of the 9 runs.

**GDA with 6 discriminator steps**

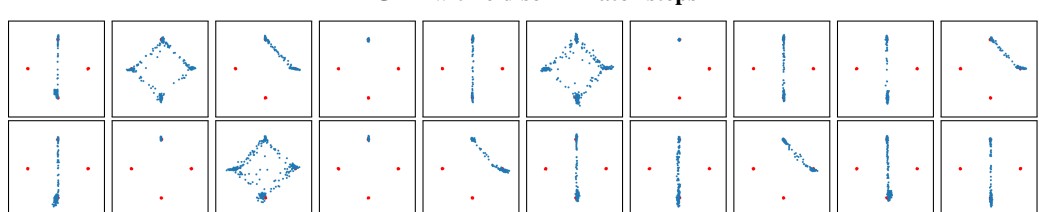

Figure 14: The generated points at the 1500'th iteration for all 20 runs of the GDA algorithm, with $k = 6$ discriminator steps, for the simulation mentioned in Figure 2. At the 1500'th iteration, GDA had learned two modes 65% of the runs, one mode 20% of the runs, and four modes 15 % of the runs.

**Unrolled GANs with 6 unrolling steps**

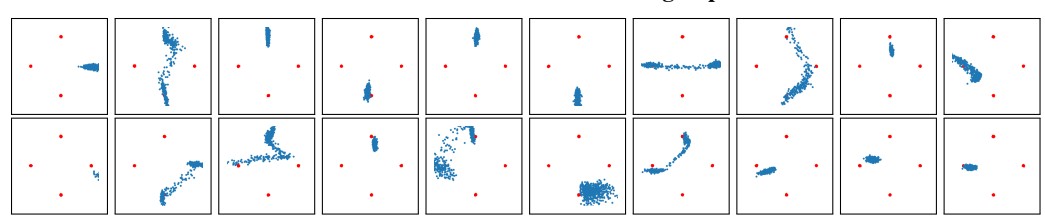

Figure 15: The generated points at the 1500'th iteration for all 20 runs of the Unrolled GAN algorithm for the example in Figure 2, with $k = 6$ unrolling steps. By the 1500'th iteration, Unrolled GANs learned one mode 75% of the runs, two modes 15% of the runs, and three modes 10% of the runs.

**Our algorithm**

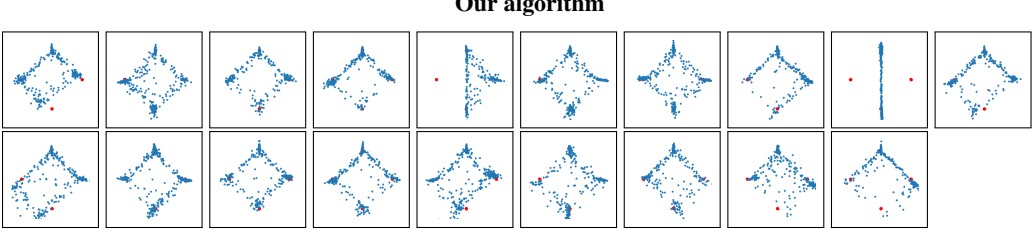

Figure 16: The generated points at the 1500'th iteration for all 19 runs of our algorithm, for the simulation mentioned in Figure 2. Our algorithm used $k = 6$ discriminator steps and an acceptance rate hyperparameter of $\frac{1}{\tau} = \frac{1}{4}$. By the 1500'th iteration, our algorithm seems to have learned all four modes 68% of the runs, three modes 26% of the runs, and two modes 5% of the runs.

## G    EXTENSION TO FUNCTIONS WITH COMPACT CONVEX SUPPORT

In this section we introduce a version of our algorithm (Section G.1) for loss functions with compact convex support, and run it on a simple bilinear loss function (Section G.2). We also introduce a version of our $\varepsilon$-local min-max equilibrium (Section G.3) which applies to loss functions with compact convex support. We then show that, if $f$ is also convex-concave, then, for $\varepsilon = 0$, this $\varepsilon$-local min-max equilibrium is equivalent to a global min-max point (Section G.4).

### G.1    PROJECTED GRADIENT MIN-MAX ALGORITHM FOR COMPACTLY SUPPORTED LOSS

---

**Algorithm 4** Algorithm for min-max optimization on compact support

---

**input:** A stochastic zeroth-order oracle $F$ for loss function $f : \mathcal{X} \times \mathcal{Y} \to \mathbb{R}$ where $\mathcal{X}, \mathcal{Y} \subseteq \mathbb{R}^d$ are compact convex sets. Stochastic gradient oracles $G_x$ for $\nabla_x f$, and $G_y$ for $\nabla_y f$. Projection oracles $\mathcal{P}_{\mathcal{X}} : \mathbb{R}^d \to \mathcal{X}$ for $\mathcal{X}$ and $\mathcal{P}_{\mathcal{Y}} : \mathbb{R}^d \to \mathcal{Y}$ for $\mathcal{Y}$. An initial point $(x, y) \in \mathcal{X} \times \mathcal{Y}$, and an error parameter $\varepsilon$.

**output:** A point $(x^\star, y^\star)$

**hyperparameters:**           $r_{\max}$      (maximum     number     of     rejections),     $\eta$       $>$ 0.

   Set $r \leftarrow 0, i \leftarrow 0$
   **while** $r \leq r_{\max}$ **do**
      $f_{\text{old}} \leftarrow F(x, y), \quad i \leftarrow i + 1$
      Set $G_x \leftarrow G_x(x, y)$ {*Compute a stochastic gradient*}
      Set $x' \leftarrow \mathcal{P}_{\mathcal{X}}(x - \eta G_x)$ {*Compute the proposed update for the min-player*}
      Starting at point $y$, use stochastic gradients $G_y(x', \cdot)$ to run multiple projected gradient ascent steps in the $y$-variable, until a point $y'$ is reached such that $\|G_y(x', y')\|_2 \leq \varepsilon$ {*Simulate max-player's update*}
      Set $j \leftarrow 0$
      Set $\mathsf{y} \leftarrow y$ and Stop = False
      **while** Stop = False **do**
         Set $j \leftarrow j + 1$
         Set $g_{\mathsf{y},j} \leftarrow G_y(\mathsf{x}, \mathsf{y}_j)$
         **if** $\frac{1}{\eta}\|\mathsf{y} - \mathcal{P}_{\mathcal{Y}}(\mathsf{y} - \eta g_{\mathsf{y},j})\|_2 > \varepsilon'$ **then**
            Set $j \leftarrow j + 1$
            Set $\mathsf{y} \leftarrow \mathcal{P}_{\mathcal{Y}}(\mathsf{y} - \eta g_{\mathsf{y},j})$ {*Simulate max-player's update via projected gradient ascent*}
         **else**
            Set Stop = True
      Set $f_{\text{new}} \leftarrow F(x', y')$ {*Compute the new loss value*}
      Set Accept $\leftarrow$ True.
      **if** $f_{\text{new}} > f_{\text{old}} - \varepsilon/2$, set Accept $\leftarrow$ False {*accept or reject*}
      **if** Accept = True **then**   Set $x \leftarrow x', y \leftarrow y', r \leftarrow 0$ {*Accept the updates*}
      **else**   Set $r \leftarrow r + 1$ {*Reject the updates, and track how many successive steps were rejected.*}
   **return** $(x, y)$

---

### G.2    NUMERICAL SIMULATION ON COMPACTLY SUPPORTED BILINEAR FUNCTION

In this appendix, we discuss numerical simulations on the bilinear loss function $f(x, y) = xy$ with compact support $(x, y) \in [-1, 1] \times [-1, 1]$. For this function, the gradient descent-ascent algorithm is known to diverge away from the global min-max point (see for instance Jin et al. (2020)).

This function has global min-max point at every point in the set $\{(x, y) : x = 0, y \in [-1, 1]\}$. We ran Algorithm 4 on this function with hyperparamters $\eta = 0.2$, $\varepsilon = 0.06$, and $r_{\max} = 5$, value oracle $F(x, y) = f(x, y)$, gradient oracle $G_y(x, y) = \nabla_y f(x, y)$, stochastic gradient oracle $\nabla_x f(x, y) + \xi$ where $\xi \sim N(0, 1)$, and initial point $(x, y) = (0.4, 0.4)$. After 341 iterations of the outer loop, our algorithm reached the point $(0.0279, -0.9944)$, which is very close to one of its true global min-max points, $(0, -1)$.

## G.3 LOCAL MIN-MAX EQUILIBRIUM FOR COMPACTLY SUPPORTED LOSS FUNCTIONS

**Global min-max point.** First, we recall the definition of global min-max point:

**Definition G.1.** *We say that $(x^\star, y^\star) \in \mathcal{X} \times \mathcal{Y}$ is a global min-max point for a function $f : \mathcal{X} \times \mathcal{Y} \to \mathbb{R}$ if*

$$f(x^\star, y^\star) = \max_{y \in \mathcal{Y}} f(x^\star, y)$$

*and*

$$f(x^\star, y^\star) = \min_{x \in \mathcal{X}} \max_{y \in \mathcal{Y}} f(x, y).$$

**Local min-max equilibrium for projected subgradients.** In this section we introduce a version of the local min-max equilibrium which applies to compactly supported loss functions (Definition G.3). The main difference with our previous definition (Definition 2.2) is the need for a projected gradient to deal with the compact support of the objective function.

In the following we assume that $f : \mathcal{X} \times \mathcal{Y} \to \mathbb{R}$, where $\mathcal{X}, \mathcal{Y} \subset \mathbb{R}^d$ are two compact convex sets, and that $f$ is continuously differentiable on $\mathcal{X} \times \mathcal{Y}$. We denote by $\nabla_{\mathrm{y}}^{\mathcal{Y}}$ the projected gradient in the $y$ variable for the set $\mathcal{Y}$.

We first formally define "simulated loss" and what it means for $f$ to increase rapidly.

**Definition G.2.** *For any $x, y$, and $\varepsilon > 0$, define $\mathcal{E}(\varepsilon, x, y) \subseteq \mathcal{Y}$ to be points $w$ s.t. there is a continuous and (except at finitely many points) differentiable path $\gamma(t)$ contained in $\mathcal{Y}$ [9], starting at $y$, ending at $w$, and moving with "speed" at most 1 in the $\ell_\infty$-norm, $\left\| \frac{\mathrm{d}}{\mathrm{d}t} \gamma(t) \right\|_\infty \leq 1$ such that at any point on $\gamma$, [10]*

$$\frac{\mathrm{d}}{\mathrm{d}t} f(x, \gamma(t)) > \varepsilon. \tag{33}$$

*We define $\mathcal{L}_\varepsilon(x, y) := \sup_{w \in \mathcal{E}(\varepsilon, x, y)} f(x, w)$, and refer to it as the simulated loss.*

**Definition G.3.** *Given a distribution $\mathcal{D}_{x,y}$, with $\Pr_{\Delta \sim \mathcal{D}_{x,y}}(x + \Delta \in \mathcal{X}) = 1$ for each $x, y \in \mathbb{R}^d$, and $\varepsilon^\star \geq 0$, we say that $(x^\star, y^\star)$ is an $\varepsilon^\star$-local min-max equilibrium with respect to the distribution $\mathcal{D}$ if*

$$\|\nabla_y^{\mathcal{Y}} f(x^\star, y^\star)\|_1 \leq \varepsilon^\star, \quad and \quad, \tag{34}$$

$$\Pr_{\Delta \sim \mathcal{D}_{x^\star, y^\star}} [\mathcal{L}_{\varepsilon^\star}(x^\star + \Delta, y^\star) < \mathcal{L}_{\varepsilon^\star}(x^\star, y^\star) - \varepsilon^\star] \leq \varepsilon^\star, \tag{35}$$

**Remark G.4.** *As a simple application of Algorithm 4, consider the bilinear loss function $f(x, y) = xy$ where $x$ and $y$ are constrained to the set $[-\frac{1}{2}, \frac{1}{2}]$. It is easy to see that the set of global min-max points consists of the points $(x, y)$ where $x = 0$ and $y$ is any point in $[-\frac{1}{2}, \frac{1}{2}]$. The local min-max equilibria according to our definition are the set of points $(x, y)$ where $x$ is any point in $[-\varepsilon, \varepsilon]$ and $y$ is any point in $[-\frac{1}{2}, \frac{1}{2}]$.*

*This is because, when running Algorithm 4 on this example, if $x$ is outside the set $[-\varepsilon, \varepsilon]$, the max-player will follow an increasing trajectory to always return a point $y = -\frac{1}{2}$ or $y = \frac{1}{2}$, which means that, roughly speaking, the min-player is attempting to minimize the function $\frac{1}{2}x$. This means that the algorithm will accept all updates $x + \Delta$ for which $\frac{1}{2}|x + \Delta| < \frac{1}{2}|x| - \frac{\varepsilon}{2}$, implying that the algorithm converges towards a point with $|x| \leq \varepsilon$.*

*Thus, as $\varepsilon$ goes to zero, the set of local min-max equilibrium points coincides with the set of global min-max optima for the function $f(x, y) = xy$.*

*This latter fact holds more generally for any convex-concave function with compact convex domain (see Theorem G.5).*

## G.4 COMPARISON OF LOCAL MIN-MAX AND GLOBAL MIN-MAX IN THE COMPACTLY SUPPORTED CONVEX-CONCAVE SETTING.

The following theorem shows that, in the compactly supported convex-concave setting, a point $(x^\star, y^\star)$ is a local min-max equilibrium for $\varepsilon = 0$ (in the sense of Definition G.3) if and only if it is a global min-max point:

---

[9]In other words there is some $\tau \geq 0$ such that $\gamma : [0, \tau] \to \mathcal{Y}$.

[10]In this equation the derivative $\frac{\mathrm{d}}{\mathrm{d}t}$ is taken from the right.

**Theorem G.5.** *Let $f : \mathcal{X} \times \mathcal{Y} \to \mathbb{R}$ be convex-concave, where $\mathcal{X}, \mathcal{Y} \subseteq \mathbb{R}^d$ are compact convex sets. And let $\mathcal{D}_{x,y}$ be a continuous distribution with support on $\mathcal{Y}$ such that, for every $(x,y) \in \mathcal{X} \times \mathcal{Y}$, $\mathcal{D}_{x,y}$ there is some open ball $B \subset \mathbb{R}^d$ containing $x$ such that $\mathcal{D}_{x,y}$ has non-zero probability density at every point in $B \cap \mathcal{Y}$. Then $(x^\star, y^\star)$ is a $\varepsilon$-local min-max equilibrium for $\varepsilon = 0$ if and only if it is a global min-max point.*

*Proof.* Define the "global max" function $\psi(x) := \max_{y \in \mathcal{Y}} f(x,y)$ for all $x \in \mathcal{X}$. We start by showing that the function $\psi(x)$ is convex on the convex set $\mathcal{X}$. Indeed, for any $x_1, x_2 \in \mathcal{X}$ and any $\lambda \in [0,1]$ we have

$$
\begin{aligned}
\lambda \psi(\lambda x_1 + (1-\lambda)x_2) &= \max_{y \in \mathcal{Y}} f(\lambda x_1 + (1-\lambda)x_2, y) \\
&\leq \max_{y \in \mathcal{Y}}[\lambda f(x_1, y) + (1-\lambda)f(x_2, y)] \\
&\leq \lambda[\max_{y \in \mathcal{Y}} f(x_1, y)] + (1-\lambda)[\max_{y \in \mathcal{Y}} f(x_2, y)] \\
&= \lambda \psi(x_1) + (1-\lambda)\psi(x_2),
\end{aligned}
$$

where the second inequality holds by convexity of $f(\cdot, y)$.

Moreover, we note that, since, for all $x \in \mathcal{X}$, $f(x, \cdot)$ is continuously differentiable on a compact convex set, every allowable path (with parameter $\varepsilon = 0$) can be extended to an allowable path whose endpoint $\hat{y}$ has projected gradient $\nabla_y^{\mathcal{Y}} f(x^\star, \hat{y}) = 0$.

Therefore, for every $(x,y) \in \mathcal{X} \times \mathcal{Y}$, there exists an allowable path with initial point $y$ whose endpoint $\hat{y}$ satisfies

$$
\nabla_y^{\mathcal{Y}} f(x, \hat{y}) = 0. \tag{36}
$$

Since $f(x, \cdot)$ is concave, equation 36 implies that

$$
f(x, \hat{y}) = \max_{y \in \mathcal{Y}} f(x, y), \tag{37}
$$

and hence that

$$
\mathcal{L}_0(x, y) = f(x, \hat{y}). \tag{38}
$$

Thus, equation 37 and equation 38 imply that

$$
\mathcal{L}_0(x, y) = \psi(x) \qquad \forall (x, y) \in \mathcal{X} \times \mathcal{Y} \tag{39}
$$

since $\psi(x) = \max_{y \in \mathcal{Y}} f(x, y)$.

1. **First we prove the "only if" direction:**

   Suppose that $(x^\star, y^\star)$ is a $\varepsilon$-local min-max equilibrium of $f$ for $\varepsilon = 0$. Let $y^\dagger$ be a global maximizer of the function $f(x^\star, \cdot)$ (the function achieves its global maximum since it is continuous and $\mathcal{Y}$ is compact). Then the projected gradient at this point is

$$
\nabla_y^{\mathcal{Y}} f(x^\star, y^\dagger) = 0. \tag{40}
$$

   Since $f(x, \cdot)$ is concave for all $x$, and $\nabla_y^{\mathcal{Y}} f(x^\star, y^\star) = 0$, at every point $y$ along the line $[y^\dagger, y^\star]$ connecting the points $y^\dagger$ and $y^\star$, equation 40 implies that

$$
\nabla_y^{\mathcal{Y}} f(x^\star, y) = 0, \qquad \forall y \in [y^\dagger, y^\star]. \tag{41}
$$

   Therefore, equation 41 implies that

$$
f(x^\star, y^\dagger) = f(x^\star, y^\star),
$$

   and hence that

$$
f(x^\star, y^\star) = \max_{y \in \mathcal{Y}} f(x^\star, y), \tag{42}
$$

   since $\max_{y \in \mathcal{Y}} f(x^\star, y) = f(x^\star, y^\dagger)$.

Now, since $(x^\star, y^\star)$ is a $\varepsilon$-local min-max equilibrium for $\varepsilon = 0$,

$$\Pr_{\Delta \sim \mathcal{D}_{x^\star, y^\star}} [\mathcal{L}_0(x^\star + \Delta, y^\star) < \mathcal{L}_0(x^\star, y^\star)] = 0. \tag{43}$$

Thus, equation 39 and equation 49 together imply that

$$\Pr_{\Delta \sim \mathcal{D}_{x^\star, y^\star}} [\psi(x^\star + \Delta) < \psi(x^\star)] = 0. \tag{44}$$

Since $\psi$ is convex, and since there is an open ball $B$ for which $\mathcal{D}_{x^\star, y^\star}$ has non-zero probability density at every point in $B \cap \mathcal{Y}$, equation 44 implies that $x^\star$ is a global minimizer for $\psi$:

$$\psi(x^\star) = \min_{x \in \mathcal{X}} \psi(x). \tag{45}$$

Therefore, equation 42 and equation 45 imply that $(x^\star, y^\star)$ is a global min-max point for $f : \mathcal{X} \times \mathcal{Y} \to \mathbb{R}$ whenever $(x^\star, y^\star)$ is a $\varepsilon$-local min-max equilibrium of $f$ for $\varepsilon = 0$.

2. **Next, we prove the "if" direction:**

Conversely, suppose that $(x^\star, y^\star)$ is a global min-max point for $f : \mathcal{X} \times \mathcal{Y} \to \mathbb{R}$. Then $f(x^\star, y^\star) = \max_{y \in \mathcal{Y}} f(x^\star, y)$. Since $f$ is differentiable on $\mathcal{X} \times \mathcal{Y}$, this implies that

$$\nabla_{\mathbf{y}}^{\mathcal{Y}} f(x^\star, y^\star) = 0. \tag{46}$$

Moreover, since $f(x^\star, y^\star)$ is a global min-max point, we also have that

$$f(x^\star, y^\star) = \min_{x \in \mathcal{X}} \left( \max_{y \in \mathcal{Y}} f(x, y) \right) = \min_{x \in \mathcal{X}} \psi(x),$$

and hence that

$$\psi(x^\star) = \min_{x \in \mathcal{X}} \psi(x). \tag{47}$$

Since we have already shown that $\psi$ is convex, equation 47 implies that

$$\Pr_{\Delta \sim \mathcal{D}_{x^\star, y^\star}} [\psi(x^\star + \Delta) < \psi(x^\star)] = 0. \tag{48}$$

Since we have also shown in equation 39 that $\psi(x) = \mathcal{L}_0(x, y)$ for all $(x, y) \in \mathcal{X} \times \mathcal{Y}$, equation 48 implies that

$$\Pr_{\Delta \sim \mathcal{D}_{x^\star, y^\star}} [\mathcal{L}_0(x^\star + \Delta, y^\star) < \mathcal{L}_0(x^\star, y^\star)] = 0. \tag{49}$$

Therefore, equation 46 and equation 49 imply that, for $\varepsilon = 0$, $(x^\star, y^\star)$ is a $\varepsilon$-local min-max equilibrium of $f : \mathcal{X} \times \mathcal{Y} \to \mathbb{R}$ whenever $(x^\star, y^\star)$ is a global min-max point for $f$.

$\square$

# H    COMPARISON TO OTHER MIN-MAX ALGORITHMS WITH MAXIMIZATION SUBROUTINES

Many algorithms for min-max games can be viewed as using an inner maximization subroutine (e.g., unrolled GANs Metz et al. (2017), and even versions of the GDA algorithm where the max-player's update is computed using multiple gradient ascent steps Goodfellow et al. (2014), as well as Maheswaranathan et al. (2019); Bolte et al. (2020)). However, in contrast to these algorithms, our algorithm has polynomial-time guarantees on the number of gradient and function evaluations when $f$ is bounded with Lipschitz Hessian.

In particular, while Bolte et al. (2020) provides theoretical guarantees, there are two key differences between their work and ours: (i) Their min-max algorithm (Algorithm 3 in their paper) requires

access to an oracle which returns the global maximum $\mathrm{argmax}_y f(x, y)$ for any input $x$. However, since $f$ can be nonconcave in $y$, computing the global maximum may be intractable and one therefore may not have access to an oracle for the global maximum value in practice. In contrast, our algorithm only requires access to a (stochastic) oracle for the gradient and function value of $f$. (ii) Bolte et al. (2020) only prove that their algorithm converges asymptotically, and do not provide any bounds on the time to convergence. In contrast, our algorithm has polynomial-time guarantees on the number of gradient and function evaluations.

## I    COMPARISON TO MIN-MAX ALGORITHMS WITH CONVEREGENCE GUARANTEES IN NONCONVEX-NONCONCAVE SETTINGS

Multiple works provide min-max optimization algorithms with convergence guarantees in various settings where $f$ may be nonconvex-nonconcave. However, the convergence results in these works still require strong assumptions on the loss function. For instance, Mertikopoulos et al. (2019); Lin et al. (2018); Gidel et al. (2019a) provide convergence guarantees under the assumption that $f$ satisfies a variational inequality, such as the "coherence" condition of Mertikopoulos et al. (2019). Specifically, one of the assumptions of this coherence condition is that there exists a global min-max solution point $(x^\star, y^\star)$ for $\min_x \max_y f(x, y)$ which satisfies the variational inequality $\langle \nabla_x f(x, y), x - x^\star \rangle - \langle \nabla_y f(x, y), y - y^\star \rangle \geq 0$ for all $(x, y)$ in $\mathbb{R}^d \times \mathbb{R}^d$. This is a relatively strong assumption since it says that at every point $(x, y) \in \mathbb{R}^d \times \mathbb{R}^d$, the vector field $(-\nabla_x f(x, y), \nabla_y f(x, y))$ must not point in a direction "away" from the global min-max point $(x^\star, y^\star)$.

Another setting where convergence guarantees have been show for min-max algorithms in the nonconvex-nonconcave setting is when $f$ satisfies a "sufficient bilinearity" condition Abernethy et al. (2019). Roughly, this condition says that there is a number $\gamma > 2$ such that, at every $x, y \in \mathbb{R}^d$, all the singular values of the cross derivative $\nabla_{xy}^2 f(x, y)$ are greater than $\gamma$. If, in addition, $f$ is 1-Lipschitz then Abernethy et al. (2019) show that their algorithm reaches a first-order $\varepsilon$-stationary point–that is a point where the gradient for the min- and max- players has magnitude at most $\varepsilon$–in roughly $O(\frac{1}{\gamma^2} \log(\frac{1}{\varepsilon}))$ evaluations of a Hessian-vector product of $f$.

In contrast to these works, our main result only assumes that the loss function is bounded with Lipschitz Hessian.

