# OpenReview forum: "A Provably Convergent and Practical Algorithm for Min-Max Optimization with Applications to GANs"
_ICLR.cc/2021/Conference — Reject_

### Official Review · AnonReviewer4 · 2020-10-27
**A Provably Convergent and Practical Algorithm for Min-Max Optimization with Applications to GANs**

**Rating:** 6
**Confidence:** 3

**Review:**

This paper proposes a new stochastic gradient descent-ascent-based method to approximate a stationary point (or local min-max solution) of a nonconvex-nonconcave minimax problem with application in GANs. The method is similar to the one in the GAN original paper, but the authors incorporate it with an acceptance rule and use a different model for the max problem. The algorithm also uses ADAM instead of standard SGD. The authors also provide some convergence guarantee to a local min-max point in polynomial-time complexity. Unfortunately, the reviewer was unable to verify the proof due to the time limit.

The reviewer finds that it is really hard to understand the proof techniques as well as the meaning of local min-max points defined in this paper especially via a neighborhood D_{x*,y*}. Many places are explained in words which are also hard to verify some of the statements. For example, Theorem 2.3 expresses the complexity in poly(...), which does not know what is the maximum order of epsilon. The proofs are also breaking into different pieces where so many technical details related to high probability statements are used. This makes another difficulty to check the correctness. To this end, the reviewer would like to raise the following question?
1. First, since the problem is nonconvex-nonconcave,  how can the algorithm guarantee that the min-play can always decrease the objective function with a certain amount that is fixed as stated in (5)?
2. Second, it is known that ADAM is not convergent even on a convex problem (see (https://openreview.net/pdf?id=ryQu7f-RZ)), do the authors use a modified variant of ADAM, or what has been changed to guarantee its convergence?
3. Third, in Definition 2, does D_{x*,y*} form a "full" neighborhood of (x*,y*) in the feasible set of f(x, y)?

---

> ### Author Response · Authors · 2020-11-11
> **Response to Review #4**
>
> Thank you for your valuable comments and suggestions.  We answer your specific questions below. If there are any other questions which would help to clarify our results, please let us know so we will have a chance to address them.
>
> “For example, Theorem 2.3 expresses the complexity in poly(...), which does not know what is the maximum order of epsilon.”
>
> In the interest of readability, we expressed the complexity in terms of poly(...) in the main body of the paper. As requested, we will clarify the explicit polynomial dependence which is $O(d^2 L^2 b^6 \varepsilon^{-11})$ for $b, L \geq 1$, and we will revise the appendix to include the explicit polynomial dependence of our result.
>
>
> “First, since the problem is nonconvex-nonconcave, how can the algorithm guarantee that the min-play can always decrease the objective function with a certain amount that is fixed as stated in (5)?”
>
> We suspect there is some confusion about the implication of Equation (5) in part 2 of our proof overview.
>
> Specifically, Equation (5) says that at the point $(x^\star, y^\star)$ where our algorithm stops, if the min-player samples an update $\Delta$ from the distribution $D$, followed by the max-player updating $y^\star$ using gradient ascent, with high probability over the sampled updates the final loss value cannot decrease by more than $\varepsilon$.
>
> We then use Equation (5) to show that the point $(x^\star, y^\star)$ where our algorithm stops is an equilibrium point.
> In case the confusion was about *when* Equation (5) holds, we would like to clarify that equation (5) only holds at the last step of our algorithm.  At intermediate steps, the min-player is able to decrease the value of the loss function with high probability.
>
> We will clarify this point in the main body of the paper.
>
> “Second, it is known that ADAM is not convergent even on a convex problem (see (https://openreview.net/pdf?id=ryQu7f-RZ)), do the authors use a modified variant of ADAM, or what has been changed to guarantee its convergence?”
>
> Yes, we use a special case of ADAM which reduces to a version of gradient ascent that converges for convex problems. In particular, for ADAM hyperparameters $\beta_1 = \beta_2 = 0$.  Our algorithm is not guaranteed to converge for arbitrary choices of ADAM hyperparameters.   We will clarify this in the paper.
>
> “Third, in Definition 2, does D_{x*,y*} form a "full" neighborhood of (x*,y*) in the feasible set of f(x, y)?”
>
> No, the distribution $D_{x^\star,y^\star}$ from which the min-player's proposed updates are sampled, may not form a full neighborhood of $x^\star$.  This is because $D_{x^\star, y^\star}$ is the distribution of the stochastic gradient noise.  The point $x^\star$ in our solution concept is a point where a random update sampled from the distribution of the stochastic gradient noise will not lead to a decrease in the loss value.
>
> While the distribution  $D_{x^\star, y^\star}$ may not always form a “full neighborhood” of $x^\star$ (in the same way that a Gaussian distribution N(0,I) would), in practice, models trained by algorithms that use stochastic gradient noise have been observed to reach minima which lead to better learning outcomes than Gaussian noise (see for instance [Zhu et al, ICML 2019 “The anisotropic noise in stochastic gradient descent: Its behavior of escaping from sharp minima and regularization effects.”])  This is the reason why we use stochastic gradient noise rather than Gaussian noise in our definition.
>
> On the other hand, please note that, for the max-player, our equilibrium point $(x^\star, y^\star)$ satisfies the condition $||\nabla_y f(x^\star, y^\star)||_1 \leq \varepsilon$. Hence, it implies that the max-player cannot take any step in a full neighborhood of $y^\star$ which increases $f$ at a rate of more than $\varepsilon$.

---

> ### Author Response · Authors · 2020-11-13
> **Revised paper**
>
> Thank you again for your helpful comments and suggestions.  In addition to the response we left below, we have also addressed your comments in the revised version of our paper. In particular, please see the note right above Theorem 2.3 which addresses your question about the ADAM hyper-parameters, the note right after equation (5) which explains the meaning of Equation (5), and the explicit polynomial dependence given in the proof of Theorem 2.3 at the top of page 23.

---

### Official Review · AnonReviewer2 · 2020-10-28
**This paper treats non-convex/non-concave min/max problems motivated by the respective problems that arise in GAN training.**

**Rating:** 6
**Confidence:** 2

**Review:**

This paper treats non-convex/non-concave min/max problems motivated by the respective problems that arise in GAN training. The main contribution is that they develop an ADAM-based algorithm that converges to \eps- local min/max points. The paper seems to be well-written and easy to follow. Moreover the proofs seem correct and sound. That said, my main concerns about this paper are twofold:

1.The idea of dividing the min/max game into minimization/maximization problems is not new (see for example: Bolte et al. (2020) A Hölderian backtracking method for min-max and min-min problems). Therefore, a reasonable question would be how this work is related with this kind of results.
2. Additionally, it is not true that Extra-Gradient are only applicable for convex-structured problems (see for example: Mertikopoulos et. al ICLR (2019), Yu Guan Hsieh et.al NeuRIPS (2019)). Hence, I think a more detailed justification towards these approaches is needed.

Overall, without being an expert myself, I would gladly raise my score if the author(s) clarify these issues in a more detailed manner.

---

> ### Author Response · Authors · 2020-11-11
> **Response to Review #2**
>
> Thank you for your valuable comments and suggestions.  We answer your specific questions below. If there are any other questions which would help to clarify our paper, please let us know so as to give us a chance to address them.
>
> 1."The idea of dividing the min/max game... for example: Bolte et al. (2020)... how this work is related with this kind of results.”
>
> Indeed, many algorithms for min-max games can be viewed as using an inner maximization subroutine (e.g., unrolled GANs [Metz et al, ICLR 2017], and even versions of the GDA algorithm where the max-player’s update is computed using multiple gradient ascent steps [Goodfellow, NeurIPS 2014], as well as the Bolte et al. (2020) work you mention).  However, in contrast to these algorithms, our algorithm has  polynomial-time guarantees on the number of gradient and function evaluations when $f$ is bounded with Lipschitz Hessian.
>
> In particular, our work has two key differences from the Bolte et al. (2020) work:
> (i) Their min-max algorithm (Algorithm 3 in their paper) requires access to an oracle which returns the global maximum $\mathrm{argmax}_y f(x,y)$ for any input $x$.  However, since $f$ can be nonconcave in $y$, computing the global maximum may be intractable and one therefore may not have access to an oracle for the global maximum value in practice.  In contrast, our algorithm only requires access to a (stochastic) oracle for the gradient and function value of $f$.
> (ii) Bolte et al. only prove that their algorithm converges asymptotically, and do not provide any bounds on the time to convergence.  In contrast, our algorithm has polynomial-time guarantees on the number of gradient and function evaluations.
>
> We will add a section that compares our algorithm to these algorithms.
>
> 2."Additionally, it is not true that Extra-Gradient are only applicable for convex-structured problems (see for example: [Mertikopoulos et. al, ICLR (2019)], [Yu Guan Hsieh et. al, NeuRIPS 2019] ). Hence, I think a more detailed justification towards these approaches is needed.”
>
> We agree that the above-mentioned papers provide convergence guarantees in settings somewhat more general than convex-structured problems. However, the convergence results in these works still require strong assumptions on the loss function. Namely, they assume that the loss function $f$ satisfies a variational inequality, such as the “coherence” condition of Mertikopoulos et. al.  ICLR (2019).
>
> Specifically, one of the assumptions of this coherence condition is that there exists a global min-max solution point $(x^\star, y^\star)$ for $\min_x \max_y f(x,y)$ which satisfies the variational inequality $\langle \nabla_x f(x,y) , x- x^\star \rangle -  \langle \nabla_y f(x,y) , y- y^\star \rangle   \geq 0$ for all $(x,y)$ in $\mathbb{R}^d \times \mathbb{R}^d$.  This is a relatively strong assumption since it says that at every point $(x,y) \in \mathbb{R}^d \times \mathbb{R}^d$,  the vector field $(- \nabla_x f(x,y),   \nabla_y f(x,y) )$  must not point in a direction “away” from the global min-max point $(x^\star, y^\star)$. In contrast, in our paper, we only assume that the loss function is bounded with Lipschitz Hessian.  We will provide a more detailed comparison to these results in a subsequent version of our paper.

---

> ### Author Response · Authors · 2020-11-13
> **Revised paper**
>
> Thank you again for your helpful comments and suggestions.  In addition to the response we left below, we have also addressed your comments in the revised version of our paper; in particular, please see Appendix H and I of the revised paper.

---

### Official Review · AnonReviewer1 · 2020-10-30

**Rating:** 4
**Confidence:** 4

**Review:**

The paper introduces a first-order algorithm for nonconvex-nonconcave min-max optimization problems. The proposed algorithm terminates in time polynomial in the dimension and smoothness parameters of the loss function. The points (x*,y*) returned by the algorithm satisfy the following guarantee: if the min-player proposes a stochastic gradient descent update to x^*, and the max-player is allowed to respond by updating y^* using any “path” that increases the loss at a rate of at least \epsilon with high probability, the final loss cannot decrease by more than \epsilon. Then the algorithm is tested in GANs settings on mixtures of Gaussians, MNIST and CIFAR-10 datasets against compared against gradient/ADAM descent ascent and Unrolled GANs. The algorithm is shown to be significantly more stable than GDA (e.g. less mode collapse, cycling, and more stable digit generation).

The paper works on the hard and ambitious problem of general non-convex non-concave optimization which has multiple AI applications such as GANs. On the proposed approach is novel considering a new solution concept and the paper provides some theoretical and experimental results. On the negative side neither the theoretical nor the experimental results seem particularly strong.

My main issue on the theoretical side has to do with the solution concept itself. The solution concept seems unnatural to me. Definition 2.1 about \El_\eps(x,y) a critical notion about the ``path" that the max-agent is allowed to use is non-constructive and obtuse. The only closely related solution concept seems to be in a recent unpublished manuscript by Mangoubi and Vishnoi. Given the novelty of the solution concept I think the authors should have spent much more time building intuition about what this concept corresponds to especially in simple settings such as bilinear zero-sum games. It seems that effectively all states satisfy the definition of the provided solution concept in a bilinear game. E.g. suppose that we are arbitrarily far from the max-min equilibrium, the min agent suggests a small improvement step now the max agent can move in the direction of the gradient for arbitrarily long distance negating any gains by the small move of the min agent. This is clearly unnatural and explains why this algorithm can terminate fast, it is because it is willing to accept arbitrarily bad states as solutions. I think that this is a major shortcoming of the solution concept. The authors seem to agree that the solution concept is rather bad at times by explicitly allowing the dynamic to escape from these points with some small probability. The theoretical analysis is definitely non-trivial but if the proposed algorithm fails to solve even simple bilinear zero-sum games then the theoretical guarantees are not particularly strong.

On the experimental side, the algorithm is being compared against weak benchmarks such as GDA. As the paper itself presents in the related work there have been a lot of recent developments on variations to the standard GDA techniques such as extra-gradient, optimistic methods, different types of averaging, etc which are known to significantly and robustly outperform GDA both theoretically and experimentally across numerous datasets. The reported FID scores are far from the state of the art and even the visual samples seem of relatively poor quality.

Overall, I believe the paper attacks a very hard problem and pursues an interesting idea but both the theoretical and experimental results seem to suggest to me that the proposed approach is not very promising.

---

> ### Author Response · Authors · 2020-11-11
> **Response to Review #1**
>
> Thank you for your valuable comments and suggestions.  We answer your specific questions below. If there are any other questions which would help to clarify our paper, please let us know so as to give us a chance to address them.
>
>
> “Definition 2.1 about \El_\eps(x,y)... “
>
> The motivation for Definition 2.1 is to model a max-player which is allowed to compute updates using a first-order algorithm.  The definition states that the max-player is allowed to use *any* smooth trajectory such that the gradient of the loss increases at a "rate" at least $\varepsilon$ along this trajectory. For example, an allowed trajectory according to Definition 2.1 is a trajectory which follows the direction of the gradient $\nabla_y f(x^\star, y)$, as long as the gradient norm $||\nabla_y f(x^\star, y)||_1$ is at least $\varepsilon$. Does this help clarify and motivate our definition?
>
> “Given the novelty of the solution concept... intuition about what this concept corresponds to especially in simple settings such as bilinear zero-sum games. ”
>
> Note that our solution concept requires that the loss function is bounded from above. For unconstrained bilinear games, the max player’s objective $\max_y f(\cdot, y)$ is infinite at almost all points and, hence, our solution concept is not applicable as is.
>
> However, it is easy to extend our solution concept for functions which are constrained to a *compact* convex domain. Specifically, one can extend our solution concept to the constrained setting by empowering the max-player to make updates via any smooth trajectory that is contained inside the domain where the norm of the *projected* gradient is at least $\varepsilon$ at every point along the trajectory.   And similarly, one can extend our algorithm to this setting by allowing the max-player to use *projected* gradient ascent.
>
> For example, consider the bilinear loss function $f(x,y) = xy$ where $x$ and $y$ are constrained to the set $[-\frac{1}{2}, \frac{1}{2}].$  It is easy to see that the set of global min-max points consists of the points $(x,y)$ where $x = 0$ and $y$ is any point in $[-\frac{1}{2}, \frac{1}{2}]$.  The local min-max equilibria according to our definition are the set of points $(x,y)$ where $x$ is any point in $[-\varepsilon, \varepsilon]$ and $y$ is any point in $[-\frac{1}{2}, \frac{1}{2}]$.
>
> This is because, when running our algorithm (modified to the constrained domain as above) on this example, if $x$ is outside the set $[-\varepsilon,\varepsilon]$, the max-player will follow an increasing trajectory to always return a point $y =  -\frac{1}{2}$ or $y = \frac{1}{2}$, which means that, roughly speaking, the min-player is attempting to minimize the function $\frac{1}{2}|x|$. This means that the algorithm will accept all updates $x+\Delta$ for which $\frac{1}{2}|x+\Delta|<  \frac{1}{2}|x| - \frac{\epsilon}{2}$, implying that the algorithm converges towards a point with $|x|\leq \varepsilon$.
>
> Thus, as $\varepsilon$ goes to zero, the set of local min-max equilibrium points coincides with the set of global min-max optima for the function $f(x,y) = xy$.
>
> This latter fact holds more generally for any convex-concave function with compact domain.
>
> We hope that this clarifies our results in the context of bilinear games.  We will include a thorough discussion on this in our paper.
>
>
> “On the experimental side...”
>
> The main contribution of this paper is to give a provably convergent algorithm for nonconvex-nonconcave min-max optimization. The point of the empirical results is to show that our algorithm leads to *stable* training of GANs (a big problem in practice), and that the per-epoch runtime and memory requirement is comparable to GDA.
>
> Our empirical results are at the same level as those in several recent GAN papers which come with some theoretical guarantees, such as (Wang et al, ICLR 2020 “On Solving Minimax Optimization Locally: A Follow-the-Ridge Approach“), and (Hsieh et al, NeurIPS 2019 “On the Convergence of Single-Call Stochastic Extra-Gradient Methods”).
>
> In addition to our comparisons to GDA, we also provide experiments which compare our algorithm to the unrolled GANs algorithm, for the problem of training a GAN on a dataset sampled from a mixture of four Gaussians.  In these experiments we observed that unrolled GANs learned three Gaussian modes 10% of the runs, two modes 15% of the runs, and one mode 75% of the runs. In contrast, our algorithm learned all four modes 68% of the runs, three modes 26% of the runs, and two modes 5% of the runs.
>
> We believe that our algorithm provides a novel and very good starting point to develop stable real-world algorithms for training GANs.

---

> > ### Comment · AnonReviewer1 · 2020-11-22
> > **Response**
> >
> > Thank you for your response to my comments. The fact that the proposed algorithm converges but does not ''solve'' f(x,y)=xy is exactly the key issue that I have with the paper. I think the proposed approach goes too far in the direction of ensuring convergence, stability to the point where it stabilizes states that clearly should not be stable. I believe this is a key issue with the proposed approach. Stability is not enough. On other hand I appreciate your responses and proposed changes. I will update my score but I remain not perfectly convinced about the current contribution.

---

> > > ### Author Response · Authors · 2020-11-22
> > > **Response**
> > >
> > > Thank you for increasing your score.   While we understand your concern, we do address it in Appendix G of our revised paper and in our previous response.
> > >
> > > Specifically, we show that our algorithm does converge to a global min-max point when applied to the loss function $f(x,y) = xy$.   In particular, please see Remark G.4 and Theorem G.5 in Appendix G, as well as the simulations in Appendix G.2, which we have added to address your concern. Could you please take another look at these sections and let us know what precisely you find lacking in our argument?

---

> > > > ### Comment · AnonReviewer1 · 2020-11-23
> > > > **Response**
> > > >
> > > > I find the fact that your original framework despite its admittedly complex set of definitions fails to apply directly in such a toy setting as the unconstrained two-dimensional setting above and needs further amendments that increase its complexity even further a negative attribute of the overall approach. I would like to have seen such drawbacks explained clearly and early on to be more supportive of the paper. The fact that such a simple example required an addendum to the paper does not make very confident that other negative examples that require further edits to the model are not possible.

---

> > > > > ### Author Response · Authors · 2020-11-24
> > > > > **Response**
> > > > >
> > > > > Thank you again for your time in reviewing the paper and for engaging in a discussion.  We emphasize that the unbounded function you mention is *not* a “negative example” to our main result.  As we have clearly stated in our original submission, our main result  (Theorem 2.3) guarantees convergence for functions whose value is *bounded* with an unconstrained domain $R^d \times R^d$.  Our main result therefore does not make any claims about unbounded functions such as the “negative example” you mention.  We note that the setting of bounded-value functions which our main result addresses is of great interest to GANs and other machine learning applications, and we will add a remark to our paper giving more concrete examples of this.
> > > > >
> > > > > Moreover, we would also like to point out that the results in Appendix G are *in addition,* and not an “edit,” to our main result. Appendix G was only added to address your question and to explain how one can extend our methods to problems beyond the important setting of bounded unconstrained functions considered in our main result.
> > > > >
> > > > > Hence, it seems that your concern does not apply to the result that is actually presented in the paper.  The fact that our methods can be adapted to address your question about unbounded functions is a positive.  We would therefore greatly appreciate if you could increase your score to reflect this.

---

> ### Author Response · Authors · 2020-11-13
> **Revised paper**
>
> Thank you again for your helpful comments and suggestions.  In addition to the response we left below, we have also addressed your comments in the revised version of our paper; in particular, please see Appendix G of the revised paper.

---

### Decision · Program_Chairs · 2021-01-07
**Final Decision**

**Decision:**

Reject

**Comment:**

All reviewers appreciated the main idea in the paper for solving the nonconvex-nonconcave minimax problems, which is deemed an extremely hard open problem. However, as R1 also pointed out, neither the theoretical nor the experimental results seem particularly strong, given that many variations of GDA and theoretical understanding of different notions of optimality have been recently developed. The paper fails to draw proper comparisons to these existing work.
Unfortunately, the paper is slightly below borderline and cannot be accepted this time.